# TRUSTED AND INTERACTIVE CLUSTERING FOR TIME-SERIES DATA

## ABSTRACT

Time-series clustering has gained abundant popularity and has been used in diverse scientific areas. However, few researchers take an information fusion perspective to combine information from the time and frequency domains to accomplish clustering, although these two domains offer distinct and complementary characteristics of time-series. Motivated by this issue, we propose a trusted and interactive model, which leverages evidence theory to combine time- and frequency-based clustering results produced by the corresponding contrastive learning module. After mathematizing clustering results from the two domains as mass functions, the uncertainty contained in these results can be quantified at the sample-specific level. The combined result thus promotes clustering reliability, and is optimized based on the pseudo-labels generated by $k$-means in an interactive learning paradigm. Both theoretical analysis and experimental results on 136 benchmark datasets validate the effectiveness of the proposed model in clustering performance. Extensive ablation experiments demonstrate the contribution of combining information from the time and frequency domains and using the interactive learning paradigm. The embeddings learned are also experimentally shown to perform well in other downstream tasks.

## 1 INTRODUCTION

Time-series clustering is an important data mining technology widely applied in different fields, such as sensor data analysis (Hayashi et al., 2024), anomaly detection (Middlehurst et al., 2024) and medical field (Zhang et al., 2024), aiming to segment time-series data samples into patterns (called clusters) with homologous characteristics (Gong et al., 2022). Unlike images, time-series data generally do not show human-recognizable features to different classes, because the label information is contained in not only the time domain but also the frequency domain (Zhang et al., 2022).

**Related work.** Most of the existing time-series clustering methods focus on the time domain, and can be divided into two categories: raw-data-based methods and feature-based ones (Ma et al., 2022). Raw-data-based methods perform clustering based on a modified similarity metric, which quantifies the distance more appropriately between time-series samples (Hayashi et al., 2024; Ferreira & Zhao, 2016; Paparrizos & Gravano, 2015; Yang & Leskovec, 2011). Feature-based methods have recently garnered greater attention, because the raw-data-based ones are not capable of modeling nonlinear temporal dependencies and multiscale (long and short-term) temporal dependencies (Ma et al., 2019a). Feature-based methods extract first informative features from raw samples in the time domain, and then the clustering algorithms are conducted on the learned features (Tang et al., 2021; Fortuin et al., 2020). Some recent works (Zhang et al., 2024; Guijo-Rubio et al., 2021) also optimize feature extraction and clustering jointly by introducing pseudo-label. For example, authors in (Péalat et al., 2023) embed the time-series onto the Stiefel manifold to obtain the geometric representations of time-series samples. STCN (Ma et al., 2022) adopts a recurrent neural network and a self-supervised clustering module, which are trained iteratively by contributing to each other. Some recent works (e.g., (Fortuin et al., 2020; Tonekaboni et al., 2022)) leverage contrastive representation learning for clustering time-series to further improve the performance. Few of the mentioned methods take an information fusion perspective to incorporate information from the time and frequency domains to accomplish clustering, although these two domains offer distinct and complementary characteristics of time-series. More detailed related work is provided in Appendix A.1.

**Motivation.** In fact, the time domain depicts the temporal evolution of signal readouts, while the frequency domain reveals the distribution of signal magnitude across different frequency components within the entire spectrum (Hyndman & Athanasopoulos, 2018). By explicitly incorporating the frequency domain, a comprehension of time-series behavior can be attained, encompassing aspects that are not fully captured by analyzing the time domain alone. Therefore, the first motivation for this work is ***incorporating frequency information to enhance the ability to detect clusters.*** Besides, the time and frequency domains can be regarded as distinct views of the same data (Cohen, 1995), and they are interconvertible through Fourier and inverse Fourier Transformation (Brigham, 1988). Given the temporal dynamics inherent in time-series data, the weights (i.e., quality) of the time and frequency domains experience fluctuations over time. Then, the second motivation is ***to dynamically describe the importance of clustering results derived from both the time and frequency domains.*** Finally, to avoid poisoning the final result with the low-quality result from one domain, the last motivation is ***to facilitate the appropriate integration of time- and frequency-based clustering results under the fast-evolving uncertain scenario.***

**Technical issues.** To address the three challenges outlined in the motivation, three technical issues must be solved.

**(1) How to develop frequency-based contrastive augmentations to obtain clustering results?** Despite the universal importance of frequency information in time-series and its pivotal role in classic signal processing (Soklaski et al., 2022), it is seldom explored in contrastive clustering for time-series data (Tonekaboni et al., 2022). This is because even a slight perturbation in the frequency domain could lead to significant alterations in the temporal patterns of the time domain (Flandrin, 1998).

**(2) How to quantify uncertainty in clustering results from the time and frequency domains?** Quantifying the uncertainty is practical for enhancing the performance of such a "two-view" time-series clustering, where the higher the uncertainty of a particular domain, the lower the weight contributing to the final result. Further, the uncertainty of every domain varies for different time-series samples.

**(3) How to formulate and optimize the fusion of clustering results with uncertainty from the time and frequency domains?** Fusion of the clustering results from the two domains belongs to the late fusion (Wang et al., 2019; Liu et al., 2018), most of which primarily cater to scenarios without any uncertainty (Wang et al., 2021). In our model, the fusion module assumes the critical responsibility of effectively managing uncertainty, all integrated seamlessly within an end-to-end framework.

**Contributions.** We introduce a trusted clustering model (named TIC) for time-series data, integrating results from the time and frequency domains within an interactive learning paradigm (shown in Fig.1). In summary, the contributions of this paper are:

- We propose to adopt novel augmentations dedicated to clustering the frequency spectrum data, through contrastive sample discrimination. This could be the first work to leverage contrastive augmentation in the frequency domain in a time-series clustering problem.

- We represent the clustering results for each sample from the time and frequency domains as two distinct mass functions (Shafer, 1976), where the sample-specific uncertainty is accurately quantified within the framework of evidence theory (DEMPSTER, 1967). The Dirichlet distribution is used to mathematize the mass function and to model the class probability distribution from each domain.

- We propose an interactive framework for time-series clustering, which could be inspiring in multiple research areas of time-series. The trusted result (obtained by combining mass functions from the time and frequency domains) and pseudo-labels (derived from k-means) contribute to each other, allowing interactive model optimization.

## 2 PRELIMINARIES: EVIDENCE THEORY

Consider a variable $\omega$ taking values in a finite set called the *frame of discernment* $\Omega = \{\omega_1, \omega_2, \ldots, \omega_C\}$, where $C$ is the number of clusters in a clustering problem. A mass function (also called a piece of evidence) (Shafer, 1976) is defined as a mapping from $2^\Omega$ to [0,1] such that

$\sum_{A \subseteq \Omega} m(A) = 1, m(A) > 0$, where $2^\Omega$ is the power set of $\Omega$ and $A$ denotes various subsets of $\Omega$. These subsets are called the *focal sets* of $m$. The value of $m(A)$ denotes a degree of belief assigned to the hypothesis "$\omega \in A$". The vacuous mass function verifies

$$m(\Omega) = m_\Omega = 1 \tag{1}$$

corresponding to total "ignorance", representing that $\omega$ may belong to any subset of the $\Omega$ (including the $\Omega$ itself). In other words, the value of $m(\Omega)$ measures the uncertainty about the values of variable $\omega$. In this work, only the singleton focal sets $\omega_1, \omega_2, \cdots, \omega_C$ and ignorance focal set $\Omega$ are considered, i.e., the mass function is in the form of $\mathbf{m} = [m(\omega_1), m(\omega_2), \cdots, m(\omega_C), m(\Omega)]$ (abbreviated as $[m_1, m_2, \cdots, m_C, m_\Omega]$).

Assume that there are 2 mass functions $m_1$ and $m_2$ on the same frame of discernment $\Omega$. The *Dempster's rule* (Shafer, 1976) used to pool the information provided by $\mathbf{m}_1$ and $\mathbf{m}_2$ is noted as $\bigoplus$ and is defined by

$$m_{1 \oplus 2}(A) = \frac{\sum_{B \cap C = A} m_{1B} m_{2C}}{\mathcal{K}_{12}}, \ A \neq \emptyset \ \& \ A \subseteq \Omega \tag{2}$$

where $B$ and $C$ are also the focal sets, and $\mathcal{K}_{12} = 1 - \sum_{B \cap C = \emptyset} m_{1B} m_{2C}$ is a normalizing factor. Dempster's rule is commutative and associative (Shafer, 1976).

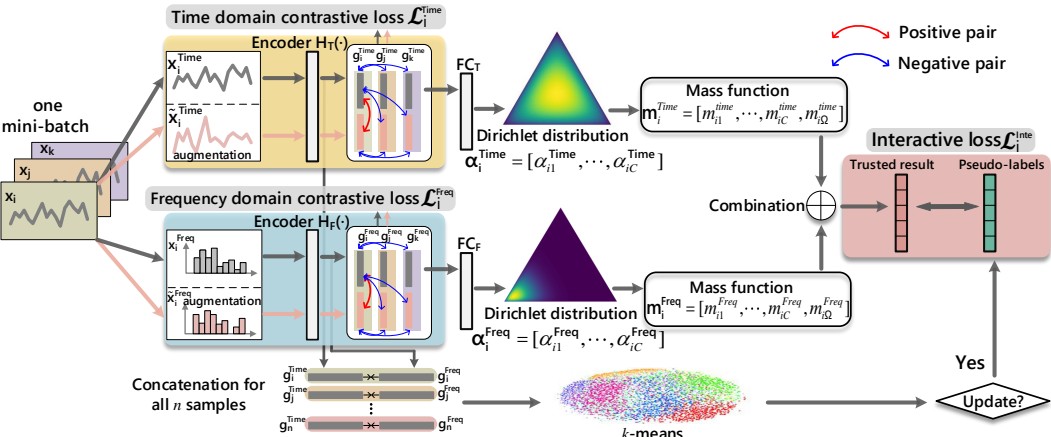

Figure 1: Overview of the proposed TIC model. TIC has three modules, a time-based contrastive module (colored yellow), a frequency-based contrastive module (colored blue), and an interactive learning module (colored red).

## 3 THE PROPOSED METHOD: TIC

**Notions and problem formulation.** We are given a time-series dataset $T = \{\mathbf{x}_1, \mathbf{x}_2, \cdots, \mathbf{x}_n\}$ of $n$ unlabeled time-series samples, and sample $\mathbf{x}_i$ has $K$ channels and $L$ time-steps. The goal is to group these $n$ samples into $C$ (a given value) clusters, which can be denoted by the *framework of discernment* $\Omega = \{\omega_1, \omega_2, \cdots, \omega_C\}$ (defined in Section 2). Without loss of generality, in the following, we focus on univariate (single-channel) time-series datasets, while noting that our TIC method can also accommodate multi-variate time-series. Superscript $\sim$ denotes contrastive augmentations, $\mathbf{x}_i \equiv \mathbf{x}_i^{\text{Time}}$ denotes the input time-series and $\mathbf{x}_i^{\text{Freq}}$ denotes the frequency spectrum of $\mathbf{x}_i$.

**Time-based contrastive module.** For the sample $\mathbf{x}_i^{\text{Time}}$ in one mini-batch, a set of augmentations $\mathcal{X}_i^{\text{Time}}$ is generated through a time-based augmentation bank $\mathcal{B}^{\text{Time}} : \mathbf{x}_i^{\text{Time}} \rightarrow \mathcal{X}_i^{\text{Time}}$ including jittering, scaling, time-shifts and other common techniques (Eldele et al., 2021). Note that the augmentations in one mini-batch are produced using diverse techniques from the augmentation bank, to expose the model to complex temporal dynamics and obtain robust embeddings.

$\mathbf{x}_i^{\text{Time}}$ and a randomly selected augmentation $\widetilde{\mathbf{x}}_i^{\text{Time}} \in \mathcal{X}_i^{\text{Time}}$ are fed into the encoder, denoted by $H_{\text{T}}(\cdot)$. The corresponding embeddings $\mathbf{g}_i^{\text{Time}} = H_{\text{T}}(\mathbf{x}_i^{\text{Time}})$ and $\widetilde{\mathbf{g}}_i^{\text{Time}} = H_{\text{T}}(\widetilde{\mathbf{x}}_i^{\text{Time}})$ are obtained.

Intuitively, embedding $\mathbf{g}_i^{\text{Time}}$ should be close to the embedding $\widetilde{\mathbf{g}}_i^{\text{Time}}$, but far away from the embeddings $\mathbf{g}_j^{\text{Time}}$ and $\widetilde{\mathbf{g}}_j^{\text{Time}}$ produced from another sample $\mathbf{x}_j^{\text{Time}}$ in the same mini-batch. Therefore, the positive pair is $(\mathbf{x}_i^{\text{Time}}, \widetilde{\mathbf{x}}_i^{\text{Time}})$ and the negative pairs are $(\mathbf{x}_i^{\text{Time}}, \mathbf{x}_j^{\text{Time}})$ and $(\mathbf{x}_i^{\text{Time}}, \widetilde{\mathbf{x}}_j^{\text{Time}})$. The normalized temperature-scaled cross-entropy loss associated with $\mathbf{x}_i^{\text{Time}}$ is defined as (Chen et al., 2020)

$$\mathcal{L}_i^{\text{Time}} = -\log \frac{\exp(\text{sim}(\mathbf{g}_i^{\text{Time}}, \widetilde{\mathbf{g}}_i^{\text{Time}}))/\tau}{\sum_{\mathbf{x}_j} \mathbb{1}_{i \neq j} \exp(\text{sim}(\mathbf{g}_i^{\text{Time}}, H_{\text{T}}(\mathbf{x}_j))/\tau)}, \tag{3}$$

where $\text{sim}(\mathbf{a}, \mathbf{b}) = \mathbf{a}^{\text{T}}\mathbf{b}/\|\mathbf{a}\|\|\mathbf{b}\|$ is the cosine similarity, $\tau$ is a temporal hyperparameter to adjust scale, $\mathbb{1}$ is an indicator function equaling to 1 when $i \neq j$ and 0 otherwise, and $\mathbf{x}_j$ denotes the sample (or its augmented sample) different from $\mathbf{x}_i$ in the same mini-batch. Minimization of $\mathcal{L}^{\text{Time}}$ enforces encoder $H_{\text{T}}(\cdot)$ to bring embeddings w.r.t. positive pairs closer together, and push embeddings w.r.t. negative pairs farther apart.

**Frequency-based contrastive module.** Although the frequency spectrum is informative, few methods leverage the frequency-based contrastive augmentation for clustering time-series (Tonekaboni et al., 2022). In this module, we use the Fourier Transformation (Brigham, 1988) to generate the frequency spectrum $\mathbf{x}_i^{\text{Freq}}$ for sample $\mathbf{x}_i^{\text{Time}}$.

As shown in (Flandrin, 1998; Zhang et al., 2022), a minor perturbation in the frequency domain can lead to significant changes in the corresponding time domain. To mitigate this issue, we manipulate the amplitude to generate frequency-based augmentation. More concretely, the augmentation bank $\mathcal{B}^{\text{Freq}} : \mathbf{x}_i^{\text{Freq}} \rightarrow \mathcal{X}_i^{\text{Freq}}$, where $\mathcal{X}_i^{\text{Freq}}$ is a set of frequency-based augmentations, includes the upgrade or downgrade of amplitude. We randomly select $\beta$ (the number of components to be manipulated) frequency components, and change each of their amplitudes from the original value $\text{Amp}_{orig}$ to $\gamma\text{Amp}_{orig}, 0 \leq \gamma < 1$ (downgrade) or to $\gamma\text{Amp}_{orig}, \gamma > 1, \text{Amp}_{orig} < \gamma\text{Amp}_{orig} \leq \text{Amp}_{\max}$ (upgrade), where $\gamma$ is a pre-defined coefficient and $\text{Amp}_{\max}$ is the maximum amplitude. Similar to the time-based contrastive module, the positive pair is $(\mathbf{x}_i^{\text{Freq}}, \widetilde{\mathbf{x}}_i^{\text{Freq}})$ and the negative pairs are $(\mathbf{x}_i^{\text{Freq}}, \mathbf{x}_j^{\text{Freq}})$ and $(\mathbf{x}_i^{\text{Freq}}, \widetilde{\mathbf{x}}_j^{\text{Freq}})$.

After generating an augmentation $\widetilde{\mathbf{x}}_i^{\text{Freq}} \in \mathcal{X}_i^{\text{Freq}}$, we feed the frequency spectrum $\mathbf{x}_i^{\text{Freq}}$ and $\widetilde{\mathbf{x}}_i^{\text{Freq}}$ into the encoder $H_{\text{F}}(\cdot)$, and obtain the embeddings $\mathbf{g}_i^{\text{Freq}}$ and $\widetilde{\mathbf{g}}_i^{\text{Freq}}$. The frequency-based loss for sample $\mathbf{x}_i^{\text{Freq}}$ is calculated as

$$\mathcal{L}_i^{\text{Freq}} = -\log \frac{\exp(\text{sim}(\mathbf{g}_i^{\text{Freq}}, \widetilde{\mathbf{g}}_i^{\text{Freq}}))/\tau}{\sum_{\mathbf{x}_j} \mathbb{1}_{i \neq j} \exp(\text{sim}(\mathbf{g}_i^{\text{Freq}}, H_{\text{F}}(\mathbf{x}_j))/\tau)}. \tag{4}$$

**Uncertainty quantification.** As shown in Fig.1, the embeddings $\mathbf{g}_i^{\text{Time}}$ and $\mathbf{g}_i^{\text{Freq}}$ are fed into the fully connected (FC) layers to obtain the clustering results in time and frequency domains. Taking the time domain as an example, the $\text{FC}_{\text{T}}$ converts the continuous embeddings to vectors $\mathbf{p}_{\text{T}} = [p_1^{\text{T}}, p_2^{\text{T}}, \cdots, p_C^{\text{T}}]$, which describes the probability of $C$ mutually exclusive events after being normalized through the *softmax* operator, and can be regarded as the parameters of a multinomial distribution (Bishop & Nasrabadi, 2006). By replacing these parameters with the parameters of a Dirichlet distribution, the clustering result from the $\text{FC}_{\text{T}}$ can be represented as a distribution over possible *softmax* outputs instead of a point estimation of one *softmax* output (Sensoy et al., 2018). That is, a Dirichlet distribution parametrized over the $\mathbf{p}_{\text{T,i}} = [p_{i1}^{\text{T}}, p_{i2}^{\text{T}}, \cdots, p_{iC}^{\text{T}}]$ represents the density of such a probability assignment w.r.t to sample $\mathbf{x}_i^{\text{Time}}$. Therefore, it can model the second-order probabilities and uncertainty for the clustering result of $\mathbf{x}_i^{\text{Time}}$ (Jsang, 2018). The definition of Dirichlet distribution is given in Definition 1.

**Definition 1** *The Dirichlet distribution is a probability density function for a categorical distribution $\boldsymbol{p}$. It can be characterized by $C$ parameters $\boldsymbol{\alpha} = [\alpha_1, \alpha_2, \cdots, \alpha_C]$ and is given by:*

$$Dir(\boldsymbol{p}|\boldsymbol{\alpha}) = \begin{cases} \frac{1}{B(\boldsymbol{\alpha})} \prod_{c=1}^{C} p_c^{\alpha_c - 1} & \text{for } \boldsymbol{p} \in \mathcal{S}_C, \\ 0 & \text{otherwise,} \end{cases} \tag{5}$$

*where $\mathcal{S}_C$ is the $C$-dimensional unit simplex $\mathcal{S}_C = \{\boldsymbol{p}| \sum_{c=1}^{C} p_c = 1, 0 \leq p_1, p_2 \cdots, p_C \leq 1\}$ and $B(\boldsymbol{\alpha})$ is the $C$-dimensional multinomial beta function.*

In the clustering problem investigated in this paper, Subjective logic (Jsang, 2018) is used to associate the parameters $\boldsymbol{\alpha} = [\alpha_1, \alpha_2, \cdots, \alpha_C]$ of a Dirichlet distribution with the output $\mathbf{p} = [p_1, p_2, \cdots, p_C]$, where the Dirichlet distribution is considered as the conjugate prior of the corresponding multinomial distribution (Bishop & Nasrabadi, 2006). Inspired from (Sensoy et al., 2018), the parameter $\alpha_c$ is calculated as

$$\alpha_c = p_c + 1. \tag{6}$$

Then, the mass function $\mathbf{m} = [m_1, m_2, \cdots, m_C, m_\Omega]$ is determined as

$$\begin{cases} m_c = \frac{p_c}{S} = \frac{\alpha_c - 1}{S} & c = 1, 2, \cdots, C, \\ \\ m_\Omega = \frac{C}{S}, \end{cases} \tag{7}$$

where $S = \sum_{c=1}^{C}(p_c + 1) = \sum_{c=1}^{C} \alpha_c$ is the Dirichlet strength (Jsang, 2018), and the value of $m_\Omega$ quantifies the uncertainty of the clustering result (as discussed in Section 2). From Eq.(7), it can be inferred that the higher value of $p_c$, the more belief mass is assigned to $m_c$. Besides, the lower value of the sum of $p_c$, the higher uncertainty $m_\Omega$ for the clustering result. For clarity, we give a specific example to explain the above formulation.

**Combining the clustering results.** After computing the mass functions $\mathbf{m}_i^{\text{Time}}$ and $\mathbf{m}_i^{\text{Freq}}$ from time and frequency domains, we use Dempster's rule to combine these two mass functions

$$\mathbf{m}_i^{\text{Comb}} = \mathbf{m}_i^{\text{Time}} \oplus \mathbf{m}_i^{\text{Freq}}, \tag{8}$$

where $\mathbf{m}_i^{\text{Comb}}$ is the combined mass function w.r.t. sample $\mathbf{x}_i$. According to Eq.(2), the specific calculation is

$$\begin{aligned} m_{ic}^{\text{Comb}} &= \frac{m_{ic}^{\text{Time}} \cdot m_{ic}^{\text{Freq}} + m_{i\Omega}^{\text{Time}} \cdot m_{ic}^{\text{Freq}} + m_{ic}^{\text{Time}} \cdot m_{i\Omega}^{\text{Freq}}}{1 - \sum_{r \neq v} m_{ir}^{\text{Time}} \cdot m_{iv}^{\text{Freq}}} \\ m_{i\Omega}^{\text{Comb}} &= \frac{m_{i\Omega}^{\text{Time}} \cdot m_{i\Omega}^{\text{Freq}}}{1 - \sum_{r \neq v} m_{ir}^{\text{Time}} \cdot m_{iv}^{\text{Freq}}}. \end{aligned} \tag{9}$$

According to Eq.(7), the combined output for $\mathbf{x}_i$ is calculated as

$$p_{ic}^{\text{Comb}} = m_{ic}^{\text{Comb}} \cdot S_i^{\text{Comb}} \text{ and } \alpha_{ic}^{\text{Comb}} = p_{ic}^{\text{Comb}} + 1, \tag{10}$$

where $S_i^{\text{Comb}} = \frac{C}{m_{i\Omega}^{\text{Comb}}}$. The time- and frequency-based clustering results are appropriately fused using Subjective logic and Dempster's rule.

**Remark 1. More intuitions: why the combined clustering results are trusted?** The produced mass value $m_{i\Omega}^{\text{Comb}}$ allows the TIC model to assess the reliability of clustering results so as to avoid risky decisions. Besides, Dempster's rule has the following advantages: (1) the mass function $\mathbf{m}^{\text{Comb}}$ obtained by combining a certain mass function (with small $m_\Omega$) with an uncertain one (with big $m_\Omega$) is still certain. This means that as long as the clustering results from either domain are trusted, then the combined clustering results are trusted, even if the results from the other domain have significant uncertainty. (2) the $\mathbf{m}^{\text{Comb}}$ obtained by combining two uncertain mass function (with large $m_\Omega$) remains uncertain. This means that if the clustering results from both the time and frequency domains are untrusted, the combined clustering results are necessarily untrusted. And users can abandon the results to avoid risks when facing this case. In order to analyze theoretically these advantages, we give the following mathematized propositions, where the combined mass function, the mass functions from the time and frequency domains are abbreviated as $\mathbf{m}^{\text{Co}}$, $\mathbf{m}^{\text{T}}$ and $\mathbf{m}^{\text{F}}$. Since $\mathbf{m}^{\text{T}}$ and $\mathbf{m}^{\text{F}}$ are equally important in the combination, the following Propositions still hold despite the exchange of the superscripts $^{\text{T}}$ and $^{\text{F}}$. The corresponding proofs are shown in Appendix A.2.

**Proposition 1** *A large $m_\Omega^{\text{T}}$ does not lead to a large $m_\Omega^{\text{Co}}$, when one of $m_c^{\text{F}}$ is large and $m_\Omega^{\text{F}}$ is small. In particular, $\mathbf{m}^{\text{Co}}$ is identical to $\mathbf{m}^{\text{T}}$, if $\mathbf{m}^{\text{F}}$ is totally uncertain (i.e., $m_\Omega^{\text{F}} = 1$).*

**Proposition 2** *The $m_\Omega^{\text{Co}}$ is monotonically increasing with $m_\Omega^{\text{T}}$ and $m_\Omega^{\text{F}}$.*

**Interactive learning module.** As shown in the lower left part of Fig.1, we concatenate the embeddings $[\mathbf{g}^{\text{Time}}; \mathbf{g}^{\text{Freq}}]$ produced from the encoders $H_{\text{T}}(\cdot)$ and $H_{\text{F}}(\cdot)$. These concatenated embeddings of $n$ samples are fed into $k$-means every $t$ epochs and update the pseudo-labels to calculate the interactive loss $\mathcal{L}^{\text{Inte}}$.

For sample $\mathbf{x}_i$, its one-hot pseudo-label vector is denoted as $\mathbf{y}_i$ with $y_{ic} = 1$ and $y_{iv} = 0$ for all $v \neq c$. In the interactive learning module, we modify the conventional cross-entropy $\mathcal{L}_i^{ce} = -\sum_{c=1}^{C} y_{ic} \log(p_{ic}^{\text{Comb}})$ as

$$\mathcal{L}_i^{'ce} = \int \left[ \sum_{c=1}^{C} -y_{ic} \log(p_{ic}) \right] \frac{1}{B(\boldsymbol{\alpha}_i)} \prod_{c=1}^{C} p_{ic}^{\alpha_{ic}-1} d\mathbf{p}_i = \sum_{c=1}^{C} y_{ic} \left( \psi(S_i) - \psi(\alpha_{ic}) \right) \quad (11)$$

where $\psi(\cdot)$ is the digamma function, $S_i$ is the Dirichlet strength w.r.t. $\mathbf{x}_i$ and we omit superscript $^{\text{Comb}}$ for brevity. Such a modification enforces the parameters $\boldsymbol{\alpha}_i^{\text{Comb}}$ of the combined Dirichlet distribution to be optimized based on the pseudo-label vector $\mathbf{y}_i$, i.e., enforces a large $p_{ic}^{\text{Comb}}$ to be produced from the $y_{ic} = 1$ in $\mathbf{y}_i$. To further shrink the $p_{iv}^{\text{Comb}}$ w.r.t. $y_{iv}$ to 0, the following KL divergence is considered

$$KL[Dir(\mathbf{p}_i|\widetilde{\boldsymbol{\alpha}}_i) \| Dir(\mathbf{p}_i|\mathbf{1})] = \log \left( \frac{\Gamma(\sum_{c=1}^{C} \widetilde{\alpha}_{ic})}{\Gamma(C) \prod_{c=1}^{C} \Gamma(\widetilde{\alpha}_{ic})} \right) + \sum_{c=1}^{C} (\widetilde{\alpha}_{ic} - 1)[\psi(\widetilde{\alpha}_{ic}) - \psi(\sum_{v=1}^{C} \widetilde{\alpha}_{iv})],$$
$$(12)$$

where $\widetilde{\boldsymbol{\alpha}}_i = \mathbf{y}_i + (1 - \mathbf{y}_i) \odot \boldsymbol{\alpha}_i$ is the adjusted parameters of Dirichlet distribution, $\mathbf{1}$ is quite a flat Dirichlet distribution and $\Gamma(\cdot)$ is the *gamma* function. Thus, the interactive loss for $\mathbf{x}_i$ is

$$\mathcal{L}_i^{\text{Inte}} = \mathcal{L}_i^{'ce} + \iota KL[Dir(\mathbf{p}_i|\widetilde{\boldsymbol{\alpha}}_i) \| Dir(\mathbf{p}_i|\mathbf{1})], \quad (13)$$

where $\iota$ is the balance hyperparameter. In summary, the sample-specific loss of TIC model is

$$\mathcal{L}_i^{\text{TIC}} = \mathcal{L}_i^{\text{Inte}} + \mathcal{L}_i^{\text{Time}} + \mathcal{L}_i^{\text{Freq}}. \quad (14)$$

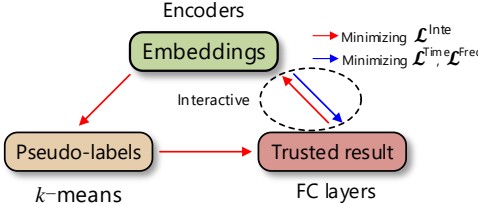

Figure 2: Illustration of the learning process in TIC. When minimizing contrastive loss $\mathcal{L}^{\text{Time}}$ and $\mathcal{L}^{\text{Freq}}$, the learned embeddings ($\mathbf{g}^{\text{Time}}$ and $\mathbf{g}^{\text{Freq}}$) affect the output of the FC layers (trusted clustering result), as shown by the blue arrow. When minimizing $\mathcal{L}^{\text{Inte}}$, the TIC model enforces the trusted clustering result to be "close" to the pseudo-labels, which are generated by inputting the embeddings into $k$-means. In the way of backward propagation, the trusted clustering result also affects the embedding learning, as shown by the red arrow. The parameter learning in FC layers and encoders $(H_{\text{T}}(\cdot), H_{\text{F}}(\cdot))$ contribute to each other interactively.

**Remark 2. More intuitions about the interactive process.** In Fig.2, we show the three main components in TIC model. The embedding learned affects the trusted clustering result when minimizing the contrastive loss $\mathcal{L}^{\text{Time}}$ and $\mathcal{L}^{\text{Freq}}$. The pseudo-labels are produced by inputting the embeddings into $k$-means and are considered when minimizing $\mathcal{L}^{\text{Inte}}$. In this way, the trusted clustering result affects the embedding learning when performing backward propagation. This allows the parameter learning of FC layers and encoders to contribute to each other interactively. Besides, considering the interactive loss avoids the class collision issue, where each sample is identified as a cluster in the embedding space (shown in Fig.3(a)) (Arora et al., 2019). This is because every positive pair consists only of the sample and its augmentation, without considering any other samples that may belong to the same cluster, when calculating the contrastive losses. The pseudo-labels in interactive loss are generated by $k$-means, where the concatenated embeddings $[\mathbf{g}^{\text{Time}}; \mathbf{g}^{\text{Freq}}]$ are treated as the input. Therefore, some basic information about clusters is included in the interactive loss and improves the clustering performance (shown in Fig.3(b)). We demonstrate quantitatively the above conclusion in Fig.4(b).

## 4 EXPERIMENTS

In this section, we conduct some experiments to answer the following research questions (RQs):

- **RQ1 (Comparison experiments)**: How does the clustering performance of TIC compare with that of other state-of-the-art time-series clustering algorithms?
- **RQ2 (Ablation study)**: How do the various components of TIC contribute to its performance?
- **RQ3 (Embedding evaluation)**: How about using the learned embedding for other downstream tasks (e.g., classification, anomaly detection)?
- **RQ4 (Hyperparameter sensitivity)**: what about the hyperparameter sensitivity of TIC?

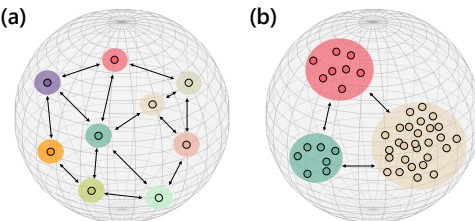

Figure 3: Illustration of the embedding space with (a) and without (b) the interactive loss. Because only the augmentation of each sample is considered when determining positive pairs, it may cause the class collision issue in the embedding space (shown in (a)), i.e., each cluster consists of only one sample (Arora et al., 2019). Optimizing the $\mathcal{L}^{\mathrm{Inte}}$ allows $k$-means to provide some basic information about the clusters and TIC to achieve better performance (shown in (b)).

**Benchmark datasets.** We use 136 benchmark (widely used in related work e.g., (Paparrizos & Gravano, 2015; Zhang et al., 2022)) time-series datasets to evaluate the TIC model. Eight datasets are collected from the real-world and the remaining 128 ones are from the UCR database (Chen et al., 2015). Eight real-world datasets are multi-variate or univariate, and their application scenarios include EEG and ECG analyses, and mechanical faulty detection. The description of these benchmark datasets is shown in Table 1. More details about the datasets are shown in Appendix A.3.

Table 1: Specific description for the 136 benchmark datasets. The detailed description of the 128 UCR datasets can be found in (Chen et al., 2015). #Sam, #Clu and #Cha denote the number of samples, clusters and channels, respectively.

| Dataset | #Sam | #Clu | #Cha | Length |
|---|---|---|---|---|
| EMG | 204 | 3 | 1 | 1,500 |
| ECG | 43,673 | 4 | 1 | 1,500 |
| HAR | 10,299 | 6 | 9 | 128 |
| Gesture | 560 | 8 | 3 | 315 |
| FD-A | 8,184 | 3 | 1 | 5,120 |
| FD-B | 13,640 | 3 | 1 | 5,120 |
| SleepEEG | 371,055 | 5 | 1 | 200 |
| Epilepsy | 11,500 | 2 | 1 | 178 |
| UCR (128 datasets) | [40, 16,637] | [2, 60] | 1 | [15, 2,844] |

**Baselines.** To answer RQ1, we consider the following 10 time-series clustering methods, i.e., **DTCR** (Ma et al., 2019b), **k-shape** (Paparrizos & Gravano, 2015), **SOM-VAE** (Fortuin et al., 2020), **STCN** (Ma et al., 2022), **TMEK** (Tang et al., 2021), **TNC** (Tonekaboni et al., 2022), **TS3C**$_{ch}$ (Guijo-Rubio et al., 2021), **UMAP** (Péalat et al., 2023), **USSL** (Zhang et al., 2019) and **VLSC** (Duan & Guo, 2023) in the comparison experiment.

To answer RQ3, we evaluate the embeddings learned by TIC on the other downstream tasks (i.e., classification and anomaly detection). The included baselines are 8 unsupervised representation learning methods, i.e., **activity2vec** (Aggarwal et al., 2019), **BTSF** (Yang & Hong, 2022), **CLOCS** (Kiyasseh et al., 2021), **MHCCL** (Meng et al., 2023), **TFC** (Zhang et al., 2022), **Triplet** (Franceschi et al., 2019), **TS2Vec** (Yue et al., 2022) and **TSTCC** (Eldele et al., 2022). More details of the baselines are shown in Appendix A.4.

**Implementation details.** In our TIC model, we use two 2-layer Transformer (Vaswani et al., 2017) as backbones for encoders $H_{\mathrm{T}}(\cdot)$ and $H_{\mathrm{F}}(\cdot)$. FC$_{\mathrm{T}}$ and FC$_{\mathrm{F}}$ contain three fully-connected layers with

hidden dimensions $d_1 = L$ (time-series length), $d_2 = 128$ and $d_3 = C$ (number of clusters), where the softmax layer is replaced with the RELU to ensure that the network outputs are non-negative values. These two FC modules do not share any parameters. The full spectrum (symmetrical) is used to guarantee that $\mathbf{x}^{\text{Time}}$ and $\mathbf{x}^{\text{Freq}}$ have the same number of dimensions, when transforming the time domain to frequency domain. The Adam optimizer with a learning rate of $\{0.0001, 0.0002, 0.0003\}$ and 2-norm penalty coefficient of 0.0005 is used. We use the batch size of $\{8, 16, 32, 64, 128\}$ according to the dataset size, and use the training epoch of 100. We set $\beta = 1$, $\gamma \in \{0.5, 1.2\}$ for the frequency augmentation and $\tau = 0.2$ in loss functions (3) and (4). The balance hyperparameter $\iota$ in Eq.(13) is gradually increased to prevent TIC model from paying too much attention to the KL divergence in the initial training stage. The pseudo-labels are updated every $t = 20$ epochs. We use the code provided in the corresponding paper, and the hyperparameters are finely tuned within the configuration provided therein based on the unsupervised metric Davies-Bouldin Index for fair comparison. The supervised ARI, NMI, ACC metrics are considered. All models are implemented with PyTorch on an NVIDIA A100 Tensor Core GPU. In Appendix A.5, the statistical comparison result derived from the Friedman test and Nemenyi test is provided.

**RQ1. Comparison experiment.** We show the comparison results between TIC and the 10 time-series clustering baselines in Table 5, where the ARI clustering metric are used (Rand, 1971). We recall that the larger the metrics, the better the clustering performance. After being fed to the correct number $C$ of clusters, each algorithm is run 5 times and the results are recorded in the form of $\text{mean}_{\pm \text{std.deviation}}$. In particular, the average ARI and the corresponding average std. deviations on the 128 UCR datasets are reported. The results w.r.t. NMI, ACC and running time are provided in Appendix A.6.

Table 2: ARI of different algorithms on benchmark datasets. The ●/○ indicates whether TIC is statistically superior/inferior to a certain comparing baseline based on the paired $t$-test at a 0.05 significance level. The statistics of win/tie/loss are shown in the last row of each sub-table. The best and the second-best results, between which the performance gaps are shown in the row named "gap" in each sub-table, are colored blue and red.

| ARI | EMG | ECG | HAR | Gesture | FD-A | FD-B | SleepEEG | Epilepsy | UCR |
|---|---|---|---|---|---|---|---|---|---|
| DTCR | $.782_{\pm.02}$● | $.812_{\pm.01}$● | $.584_{\pm.02}$● | $.874_{\pm.03}$● | $.882_{\pm.02}$● | $.801_{\pm.01}$● | $.571_{\pm.01}$● | $.872_{\pm.02}$● | $.634_{\pm.02}$● |
| $k$-shape | $.684_{\pm.02}$● | $.578_{\pm.01}$● | $.765_{\pm.03}$ | $.723_{\pm.02}$● | $.860_{\pm.01}$● | $.811_{\pm.02}$● | $.806_{\pm.01}$● | $.909_{\pm.02}$● | $.802_{\pm.03}$● |
| SOM-VAE | $.845_{\pm.03}$● | $.812_{\pm.02}$● | $.774_{\pm.01}$ | $.921_{\pm.01}$ | $.824_{\pm.03}$● | $.732_{\pm.02}$● | $.816_{\pm.02}$● | $.931_{\pm.01}$● | $.813_{\pm.02}$● |
| STCN | $1_{\pm0}$ | $.730_{\pm.02}$● | $.669_{\pm.02}$● | $.825_{\pm.01}$● | $.879_{\pm.02}$● | $.803_{\pm.01}$● | $.815_{\pm.02}$● | $.770_{\pm.01}$● | $.541_{\pm.02}$● |
| TMEK | $.651_{\pm.02}$● | $.706_{\pm.02}$● | $.690_{\pm.01}$● | $.807_{\pm.01}$● | $.682_{\pm.02}$● | $.782_{\pm.01}$● | $.741_{\pm.02}$● | $.901_{\pm.02}$● | $.642_{\pm.01}$● |
| TNC | $.751_{\pm.01}$● | $.805_{\pm.02}$● | $.693_{\pm.01}$● | $.851_{\pm.02}$● | $.780_{\pm.03}$● | $.593_{\pm.01}$● | $.741_{\pm.02}$● | $.725_{\pm.01}$● | $.707_{\pm.01}$● |
| TS3C$_{ch}$ | $.703_{\pm.02}$● | $.813_{\pm.02}$● | $.706_{\pm.01}$● | $.852_{\pm.01}$● | $.711_{\pm.02}$● | $.682_{\pm.01}$● | $.865_{\pm.01}$ | $.939_{\pm.02}$● | $.851_{\pm.02}$ |
| UMAP | $.859_{\pm.01}$● | $.791_{\pm.01}$● | $.643_{\pm.02}$● | $.852_{\pm.02}$● | $.725_{\pm.01}$● | $.785_{\pm.01}$● | $.863_{\pm.01}$ | $.972_{\pm.02}$ | $.848_{\pm.01}$● |
| USSL | $.876_{\pm.01}$● | $.745_{\pm.01}$● | $.621_{\pm.02}$● | $.597_{\pm.02}$● | $.622_{\pm.03}$● | $.592_{\pm.03}$● | $.802_{\pm.02}$● | $.925_{\pm.02}$● | $.710_{\pm.01}$● |
| VLSC | $1_{\pm0}$ | $.805_{\pm.01}$● | $.611_{\pm.02}$● | $.823_{\pm.02}$● | $.654_{\pm.02}$● | $.498_{\pm.01}$● | $.796_{\pm.01}$● | $.942_{\pm.02}$ | $.757_{\pm.01}$● |
| TIC (ours) | $1_{\pm0}$ | $.879_{\pm.01}$ | $.796_{\pm.01}$ | $.931_{\pm.01}$ | $.939_{\pm.01}$ | $.853_{\pm.02}$ | $.896_{\pm.01}$ | $.980_{\pm.01}$ | $.883_{\pm.01}$ |
| gap | .124 | .066 | .022 | .010 | .057 | .042 | .031 | .008 | .032 |
| win/tie/loss | 8/2/0 | 10/0/0 | 8/2/0 | 9/1/0 | 10/0/0 | 10/0/0 | 8/2/0 | 8/2/0 | 9/1/0 |

Overall, our TIC model wins 80 and has tied performance on 10 out of 90 trials, when it is statistically compared with 10 baselines based on three metrics. On average, our TIC model claims a large performance gap of 0.043 over the best baselines. Concretely, the largest performance gap is 0.0124 on the EMG dataset and the smallest one is 0.008 on the Epilepsy dataset. Only on the EMG dataset, STCN and VLSC yield the totally correct clustering result, and achieve the equal ARI with TIC. One potential explanation is that EMG is a simple dataset with only 3 clusters and 204 samples, and thus it may be easy to be modeled. On the ECG dataset, TIC outperforms the strongest baselines by a large margin of 0.066. Because ECG includes four clusters consisting of 43,673 samples that lead to a more complex clustering task, a combination of clustering information in the time and frequency domains results in better performance. In summary, the better performance of TIC model can be attributed to (1) Both time-domain and frequency-domain contrastive losses are considered; (2) Quantification of sample-specific uncertainty reduces the wrong assignment in clustering decision; (3) Dempster's rule appropriately combines the time- and frequency-based clustering results; (4) the class collision issue can be improved by including the interactive loss $\mathcal{L}^{\text{Inte}}$ in $\mathcal{L}^{\text{TIC}}$. The ablation study (more results shown in Appendix A.7) experimentally demonstrates the four points above.

**RQ2. Ablation study.** We conduct ablation studies to evaluate the importance of every component in the developed TIC model. Concretely, we compare TIC model with its 3 variants: W/o $\mathcal{L}^{\text{Time}}/\mathcal{L}^{\text{Freq}}$: the loss function $\mathcal{L}^{\text{Time}}/\mathcal{L}^{\text{Freq}}$ is removed from the $\mathcal{L}^{\text{TIC}}$, i.e., the time-

based/frequency-based contrastive module is removed; W/o $\mathcal{L}^{\mathrm{Inte}}$: the loss function $\mathcal{L}^{\mathrm{Inte}}$ is removed from the $\mathcal{L}^{\mathrm{TIC}}$, i.e., $k$-means does not provide pseudo-labels for the training.

Table 3: ARI values (mean$_{\pm \mathrm{std.deviation}}$) of different variants of TIC.

| ARI | EMG | ECG | HAR | Gesture | FD-A | FD-B | SleepEEG | Epilepsy | UCR | Average |
|---|---|---|---|---|---|---|---|---|---|---|
| W/o $\mathcal{L}^{\mathrm{Time}}$ | $.958_{\pm.01}$ | $.815_{\pm.02}$ | $.702_{\pm.01}$ | $.846_{\pm.02}$ | $.859_{\pm.02}$ | $.745_{\pm.02}$ | $.687_{\pm.02}$ | $.899_{\pm.01}$ | $.756_{\pm.01}$ | 0.807 |
| W/o $\mathcal{L}^{\mathrm{Freq}}$ | $.934_{\pm.01}$ | $.827_{\pm.01}$ | $.754_{\pm.01}$ | $.798_{\pm.02}$ | $.547_{\pm.02}$ | $.609_{\pm.01}$ | $.781_{\pm.02}$ | $.854_{\pm.01}$ | $.716_{\pm.01}$ | 0.758 |
| W/o $\mathcal{L}^{\mathrm{Inte}}$ | $.042_{\pm.03}$ | $.121_{\pm.02}$ | $.098_{\pm.01}$ | $.057_{\pm.03}$ | $.069_{\pm.02}$ | $.210_{\pm.01}$ | $.009_{\pm.02}$ | $.147_{\pm.02}$ | $.059_{\pm.01}$ | 0.090 |
| TIC (Full model) | $\mathbf{1}_{\pm 0}$ | $\underline{\mathbf{.879}}_{\pm.01}$ | $\underline{\mathbf{.796}}_{\pm.01}$ | $\underline{\mathbf{.931}}_{\pm.01}$ | $\underline{\mathbf{.939}}_{\pm.01}$ | $\underline{\mathbf{.853}}_{\pm.02}$ | $\underline{\mathbf{.896}}_{\pm.01}$ | $\underline{\mathbf{.980}}_{\pm.01}$ | $\underline{\mathbf{.883}}_{\pm.01}$ | **0.906** |

The results of the ablation study are reported in Table 3. As can be seen, removing $\mathcal{L}^{\mathrm{Time}}$ and $\mathcal{L}^{\mathrm{Freq}}$ leads to performance degradation (average ARI) of $0.912 - 0.807 = 0.105$ and $0.912 - 0.758 = 0.154$, respectively. In particular, on the datasets {FD-A, FD-B} with a high sampling frequency of 64k Hz, removing $\mathcal{L}^{\mathrm{Freq}}$ cause more pronounced performance degradation, e.g., the ARI values of W/o $\mathcal{L}^{\mathrm{Time}}$ and $\mathcal{L}^{\mathrm{Freq}}$ are 0.859 and 0.547 on FD-A dataset. One can conclude that the time- and frequency-based contrastive modules have almost the same contribution to the whole TIC model. The variant W/o $\mathcal{L}^{\mathrm{Inte}}$ has the lowest ARI on all the datasets. This is because every positive pair consists only of the sample and its augmentation after removing the $\mathcal{L}^{\mathrm{Inte}}$, i.e., losing the basic clustering information provided by pseudo-labels. In this case, the class collision issue seriously degrades the clustering performance as discussed in Remark 2. In Fig.4(b), we further show the cosine distances among samples belonging to different clusters. One can see that considering the interactive loss $\mathcal{L}^{\mathrm{Inte}}$ (with $\mathcal{L}^{\mathrm{Inte}}$ in Fig.4(b)) can increase the inter-cluster cosine distance with a remarkable gap of 20.4% (i.e., from 1.965 to 2.365 on ECG dataset). It shows that minimizing the $\mathcal{L}^{\mathrm{Inte}}$ indeed increases the distinctiveness of learned embeddings and bring forward better cluster performance.

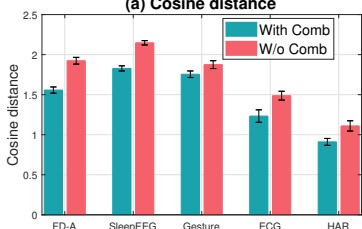 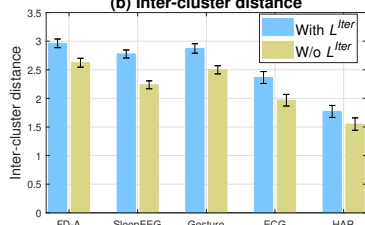

Figure 4: Cosine distance (a) between time- and frequency-based embeddings of the same sample considering the combination (With comb, i.e., full TIC model) and without the combination (W/o Comb) of results from time and frequency domains. The lower distance denotes better encoder learning, because $\mathbf{g}_i^{\mathrm{time}}$ and $\mathbf{g}_i^{\mathrm{time}}$ are closer in embedding space. Cosine distance (b) among the concatenated embeddings $[\mathbf{g}_i^{\mathrm{time}};\mathbf{g}_i^{\mathrm{time}}]$ of samples belonging to different clusters. The larger inter-cluster distance represents a better clustering result.

**RQ3. Embedding evaluation.** We evaluate the embedding learned by TIC model by performing two other downstream tasks: classification and anomaly detection. The results w.r.t. anomaly detection are shown in Appendix A.8. We follow the same protocol as (Franceschi et al., 2019; Tonekaboni et al., 2022), where a multi-class SVM with RBF kernel and a linear classifier are trained on top of the embeddings learned by different models. The 5-fold cross-validation is adopted to train the SVM and the linear classifier. In TIC, the time- and frequency-based embeddings are concatenated $[\mathbf{g}^{\mathrm{Time}};\mathbf{g}^{\mathrm{Freq}}]$. Beyond the aforementioned unsupervised representation learning methods, we consider a K-nearest neighbor classifier ($K = 5$) equipped with DTW metric (KNN$_{\mathrm{DTW}}$) as another baseline. The evaluation results are summarized in Table 4, where the results w.r.t the linear classifier and SVM are shown in the first and second sub-tables. As can be seen, TIC achieves the best performance (colored blue) in 13 out of 18 cases and the second-best performance in another 5 cases. In particular, TIC shows the highest accuracy of 0.9654 when training an SVM classifier, which yields a margin of 4.3% over the best baseline activity2vec (0.9257). It shows that TIC adequately leverages the information from time and frequency domains to provide more fine-grained embeddings for discriminant. Besides, almost all the unsupervised representation learning baselines have higher accuracy than KNN$_{\mathrm{DTW}}$, illustrating the dominance of neural networks in representation learning.

Table 4: Accuracy (mean$_{\pm \text{std.deviation}}$) of different methods on benchmark datasets. The results w.r.t the linear classifier and the multi-class SVM are shown in the first and second sub-tables. KNN$_{\text{DTW}}$ denotes a K-nearest neighbor classifier equipped with DTW metric.

| Linear | EMG | ECG | HAR | Gesture | FD-A | FD-B | SleepEEG | Epilepsy | UCR |
|---|---|---|---|---|---|---|---|---|---|
| KNN$_{\text{DTW}}$ | .8452$_{\pm.089}$ | .6712$_{\pm.053}$ | .7152$_{\pm.065}$ | .5746$_{\pm.031}$ | .6784$_{\pm.053}$ | .3462$_{\pm.078}$ | .6745$_{\pm.043}$ | .8756$_{\pm.074}$ | .7127$_{\pm.054}$ |
| activity2vec | .9587$_{\pm.011}$ | .7853$_{\pm.036}$ | .9258$_{\pm.073}$ | .6547$_{\pm.026}$ | **.8964**$_{\pm.039}$ | .6578$_{\pm.038}$ | .8941$_{\pm.085}$ | .9245$_{\pm.004}$ | **.9123**$_{\pm.030}$ |
| BTSF | .9578$_{\pm.057}$ | .8514$_{\pm.034}$ | **.9463**$_{\pm.068}$ | **.8637**$_{\pm.079}$ | .8875$_{\pm.068}$ | .7123$_{\pm.029}$ | .8745$_{\pm.075}$ | .9521$_{\pm.056}$ | .8654$_{\pm.028}$ |
| CLOCS | .8924$_{\pm.049}$ | .7546$_{\pm.056}$ | .8954$_{\pm.038}$ | .4852$_{\pm.023}$ | .8456$_{\pm.087}$ | .7214$_{\pm.036}$ | .8875$_{\pm.074}$ | .9485$_{\pm.029}$ | .8745$_{\pm.054}$ |
| MHCCL | .9643$_{\pm.022}$ | .7765$_{\pm.031}$ | .9164$_{\pm.034}$ | .7756$_{\pm.055}$ | .8382$_{\pm.064}$ | .7936$_{\pm.051}$ | **.9145**$_{\pm.029}$ | .9654$_{\pm.005}$ | .7363$_{\pm.098}$ |
| TFC | **.9854**$_{\pm.007}$ | **.9087**$_{\pm.045}$ | .9245$_{\pm.054}$ | .7955$_{\pm.024}$ | .8657$_{\pm.048}$ | **.8834**$_{\pm.038}$ | .8921$_{\pm.051}$ | .9478$_{\pm.025}$ | .7982$_{\pm.069}$ |
| Triplet | .9338$_{\pm.077}$ | .8937$_{\pm.064}$ | .9054$_{\pm.029}$ | .6953$_{\pm.044}$ | .8542$_{\pm.055}$ | .8377$_{\pm.047}$ | .8999$_{\pm.035}$ | **.9785**$_{\pm.009}$ | .8032$_{\pm.013}$ |
| TS2Vec | .8687$_{\pm.035}$ | .8546$_{\pm.054}$ | .4887$_{\pm.067}$ | .8365$_{\pm.081}$ | .8922$_{\pm.039}$ | .8643$_{\pm.059}$ | .8456$_{\pm.062}$ | .9087$_{\pm.029}$ | .8266$_{\pm.002}$ |
| TSTCC | .9485$_{\pm.014}$ | .7481$_{\pm.011}$ | .8804$_{\pm.025}$ | .7685$_{\pm.024}$ | .8547$_{\pm.039}$ | .8598$_{\pm.011}$ | .8300$_{\pm.007}$ | .9158$_{\pm.009}$ | .7591$_{\pm.003}$ |
| TIC (ours) | .9785$_{\pm.057}$ | **.9132**$_{\pm.056}$ | **.9571**$_{\pm.084}$ | **.8795**$_{\pm.067}$ | **.9215**$_{\pm.069}$ | **.8965**$_{\pm.027}$ | **.9013**$_{\pm.039}$ | **.9874**$_{\pm.055}$ | **.9251**$_{\pm.036}$ |
| SVM | EMG | ECG | HAR | Gesture | FD-A | FD-B | SleepEEG | Epilepsy | UCR |
| KNN$_{\text{DTW}}$ | .8452$_{\pm.089}$ | .6712$_{\pm.053}$ | .7152$_{\pm.065}$ | .5746$_{\pm.031}$ | .6784$_{\pm.053}$ | .3462$_{\pm.078}$ | .6745$_{\pm.043}$ | .8756$_{\pm.074}$ | .7127$_{\pm.054}$ |
| activity2vec | .9651$_{\pm.035}$ | .8324$_{\pm.067}$ | **.9257**$_{\pm.054}$ | .6871$_{\pm.036}$ | .9031$_{\pm.089}$ | .6982$_{\pm.056}$ | .8874$_{\pm.074}$ | .9483$_{\pm.036}$ | **.9245**$_{\pm.057}$ |
| BTSF | .9687$_{\pm.045}$ | .8789$_{\pm.037}$ | .9128$_{\pm.061}$ | **.8843**$_{\pm.039}$ | .8992$_{\pm.066}$ | .7536$_{\pm.052}$ | .8905$_{\pm.037}$ | .9569$_{\pm.045}$ | .8763$_{\pm.079}$ |
| CLOCS | .9214$_{\pm.066}$ | .7895$_{\pm.074}$ | .9214$_{\pm.056}$ | .5123$_{\pm.081}$ | .8517$_{\pm.036}$ | .7541$_{\pm.063}$ | .8907$_{\pm.046}$ | **.9681**$_{\pm.039}$ | .8852$_{\pm.028}$ |
| MHCCL | .9695$_{\pm.035}$ | .7854$_{\pm.076}$ | .9237$_{\pm.046}$ | .7986$_{\pm.048}$ | .8736$_{\pm.078}$ | .8065$_{\pm.031}$ | **.8943**$_{\pm.032}$ | .9533$_{\pm.067}$ | .7629$_{\pm.054}$ |
| TFC | **.9743**$_{\pm.085}$ | **.9316**$_{\pm.037}$ | .9158$_{\pm.045}$ | .8214$_{\pm.061}$ | .8702$_{\pm.047}$ | **.9317**$_{\pm.036}$ | .8795$_{\pm.026}$ | .9538$_{\pm.054}$ | .8369$_{\pm.091}$ |
| Triplet | .9502$_{\pm.045}$ | .9125$_{\pm.067}$ | .9056$_{\pm.057}$ | .8506$_{\pm.075}$ | .8722$_{\pm.049}$ | .8697$_{\pm.094}$ | .8907$_{\pm.056}$ | .9654$_{\pm.048}$ | .8537$_{\pm.033}$ |
| TS2Vec | .8794$_{\pm.024}$ | .8874$_{\pm.079}$ | .5638$_{\pm.047}$ | .8574$_{\pm.067}$ | **.9201**$_{\pm.043}$ | .8932$_{\pm.046}$ | .8645$_{\pm.033}$ | .9268$_{\pm.076}$ | .8437$_{\pm.070}$ |
| TSTCC | .9542$_{\pm.036}$ | .7896$_{\pm.033}$ | .9214$_{\pm.096}$ | .7982$_{\pm.019}$ | .8964$_{\pm.056}$ | .8609$_{\pm.087}$ | .8514$_{\pm.076}$ | .9235$_{\pm.044}$ | .7978$_{\pm.021}$ |
| TIC (ours) | **.9855**$_{\pm.065}$ | **.9236**$_{\pm.049}$ | **.9654**$_{\pm.037}$ | **.9025**$_{\pm.081}$ | **.9187**$_{\pm.064}$ | **.9222**$_{\pm.074}$ | **.9098**$_{\pm.026}$ | **.9745**$_{\pm.037}$ | **.9391**$_{\pm.056}$ |

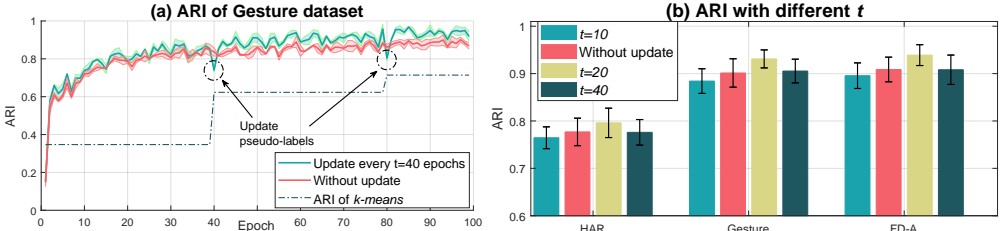

Figure 5: ARI in every epoch with updating pseudo-labels every 40 epochs and without update (a). ARI decreases slightly each time the pseudo-labels are updated by $k$-means at $40^{th}$ and $80^{th}$ epochs (shown by black circles). ARI with different $t$ is shown in (b). The frequent updating of pseudo-labels ($t = 10$) degrades the clustering performance of TIC model.

**RQ4. Sensitivity of the hyperparameter** $t$. As shown in Fig.1, TIC model updates the pseudo-labels generated by $k$-means every $t$ epochs. In Fig.5(a), we show the ARI of TIC in every epoch with $t = 40$ and without update for the Gesture dataset. It can be seen that the clustering results generated by $k$-means keep being improved (ARI=0.347 with epoch $\in [1, 39]$, ARI=0.623 with epoch $\in [40, 79]$, ARI=0.714 with epoch $\in [80, 99]$) because the embedding learning is constantly optimized. This leads to basic clustering information of higher quality being considered in the interaction loss. Although the ARI of TIC decreases temporarily at each update of the pseudo-label (epoch $= 40$ and epoch $= 80$), the final ARI with $t = 40$ is higher than the one without updates. Fig.5(b) shows the ARI of TIC model with different $t$. Frequent updating ($t = 10$) of pseudo-labels degrades the performance of TIC, even making it lower than the case of no updating, as the embedding learning is destabilized. we also provide the visualization of the clustering results and the effects of various data augmentation techniques in Appendix A.9 and A.10.

## 5 CONCLUSION

This paper proposes a trusted and interactive clustering model for time-series data, named TIC, leveraging evidence theory to combine time- and frequency-based information. TIC optimizes the contrastive loss from time and frequency domains, and an interactive loss calculated based on the pseudo-labels. Uncertainty in time- and frequency-based clustering results are quantified by mass functions that are combined by Dempster's rule to produce the trusted clustering results. Experimental results show the superior clustering performance of TIC brought by the combination of clustering results from time and frequency domains, as well as the consideration of interactive loss. The embedding learned by TIC is also shown to perform well on the classification and anomaly detection tasks.

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

# A  APPENDIX

## A.1  RELATED WORK

We review time-series clustering methods, contrastive learning methods toward time-series data and uncertain learning methods, respectively.

### A.1.1  TIME-SERIES CLUSTERING

Time-series clustering methods can be roughly classified into two families: raw-data-based methods and feature-based methods.

**Raw-data-based methods.** These methods primarily modify the distance metric to accommodate the specific characteristics of time-series data. K-DBA (Petitjean et al., 2011) combines $k$-means and dynamic time warping (DTW) (Itakura, 1975) to achieve improved alignment. To reveal the temporal dynamics, K-SC method (Yang & Leskovec, 2011) utilizes a similarity metric that is invariant to scaling and shifting. K-Shape (Paparrizos & Gravano, 2015) considers the shapes of time-series by employing a normalized cross-correlation metric. The mentioned methods often exhibit sensitivity to outliers and noise because they consider all time-step points (Ma et al., 2019b).

**Feature-based methods.** Feature-based methods typically involve two stages, where the input time-series samples are transformed into informative features first and clustering algorithms are then conducted on these features (Tang et al., 2021; Fortuin et al., 2020). In conjunction with $k$-means, TNC (Tonekaboni et al., 2022) ensures the distribution of signals from the neighborhood is distinguishable from the distribution of non-neighboring signals. Authors in (Tang et al., 2021) map the raw time-series space into multiple kernel spaces via elastic distance measure functions and resort to a self-paced learning paradigm to group time-series samples. Authors in (Péalat et al., 2023) embed the time-series onto the Stiefel manifold to obtain the geometric representations of time-series samples. STCN (Ma et al., 2022) optimizes the feature extraction and clustering simultaneously, through a recurrent neural network and a self-supervised clustering module. However, almost all the feature-based methods do not adopt an information fusion perspective to incorporate both time and frequency domain information for the purpose of clustering.

### A.1.2  CONTRASTIVE LEARNING TOWARD TIME-SERIES DATA

Contrastive learning is a well-known form of self-supervised learning and aims to train an encoder that maps original inputs into an embedding space. The objective is to bring positive sample pairs closer together, while pushing negative sample pairs (comprising the original augmentation and an alternative augmentation of a different input sample) apart (Chen et al., 2020). Compared to CV (Changpinyo et al., 2021) and NLP (Devlin et al., 2018), contrastive learning in the context of time-series data has been less explored, mainly due to the difficulty of capturing crucial invariance properties specific to time-series. TF-C (Zhang et al., 2022) expects that time-based and frequency-based representations of the same sample are located close together in the time-frequency space, and embeds the time-based neighborhood of a sample close to its frequency-based neighborhood. BTSF (Yang & Hong, 2022) utilizes sample-level augmentation with a dropout on a time-series sample, and devises the iterative bilinear temporal-spectral fusion to generate discriminative embeddings. CoST (Woo et al., 2021) comprises both time domain and frequency domain contrastive losses to learn seasonal representations for long sequence time-series forecasting. In addition to these three methods involving both time and frequency domains, other methods mainly focus on the augmentations implemented in time domain, such as transformation invariance (e.g., SimCLR (Tang et al., 2020; Chen et al., 2020)) and contextual invariance (e.g., TS2vec (Yue et al., 2022) and TS-TCC

(Eldele et al., 2021)). In previous works, the loss information from the time- and frequency-domain is captured in a compositional way, e.g., TF-C simply sums the loss functions from the two domains and the consistency loss function, and BTSF solely implements the data augmentation in the time domain. To the best of our knowledge, this work is the first one that directly combines time-frequency domain information to leverage information fusion for time-series clustering.

### A.1.3 Uncertainty-based learning

Some efforts (Gal & Ghahramani, 2016; Lakshminarayanan et al., 2017; Charpentier et al., 2020) have been made to enable the neural network to estimate the uncertainty of the output. Evidential network (Sensoy et al., 2018) incorporates subjective logic to model the Dirichlet distribution. Post-Net (Malinin et al., 2019) utilizes normalizing flow and Bayesian loss during training to estimate uncertainty. TMC (Han et al., 2021) and ETMC (Han et al., 2022) introduce a variational Dirichlet distribution to characterize the distribution of the class probabilities in multi-view classification. Ensemble distribution distillation (Malinin et al., 2019) leverages the predictions of multiple models to estimate the uncertainty. Authors in (Kopetzki et al., 2021) apply median smoothing to the Dirichlet model and enhance the capability of the model to handle adversarial examples. Unlike the above methods, our method is perhaps the first attempt to estimate the uncertainty of the outputs from a neural network oriented to the clustering problem. Further, we fuse the output with uncertainty estimation from the time and frequency domains to obtain trusted clustering results under the framework of evidence theory.

### A.2 Proofs of Propositions 1 and 2

**Proposition 1:** *A large $m_\Omega^{\mathrm{T}}$ does not lead to a large $m_c^{\mathrm{Co}}$, when one of $m_c^{\mathrm{F}}$ is large and $m_\Omega^{\mathrm{F}}$ is small. In particular, $\boldsymbol{m}^{\mathrm{Co}}$ is identical to $\boldsymbol{m}^{\mathrm{T}}$, if $\boldsymbol{m}^{\mathrm{F}}$ is totally uncertain (i.e., $m_\Omega^{\mathrm{F}} = 1$).*

*Proof:*

$$
\begin{aligned}
m_c^{\mathrm{Co}} &= \frac{m_c^{\mathrm{T}} m_\Omega^{\mathrm{F}} + m_c^{\mathrm{F}} m_\Omega^{\mathrm{T}} + m_c^{\mathrm{T}} m_c^{\mathrm{F}}}{m_\Omega^{\mathrm{F}} m_\Omega^{\mathrm{T}} + \sum_{v=1}^{C} m_v^{\mathrm{T}} m_v^{\mathrm{F}} + (1 - m_\Omega^{\mathrm{T}}) m_\Omega^{\mathrm{F}} + (1 - m_\Omega^{\mathrm{F}}) m_\Omega^{\mathrm{T}}} \\
&= \frac{m_c^{\mathrm{T}} m_\Omega^{\mathrm{F}} + m_c^{\mathrm{F}} m_\Omega^{\mathrm{T}} + m_c^{\mathrm{T}} m_c^{\mathrm{F}}}{\sum_{v=1}^{C} m_v^{\mathrm{T}} m_v^{\mathrm{F}} + m_\Omega^{\mathrm{T}} + m_\Omega^{\mathrm{F}} - m_\Omega^{\mathrm{F}} m_\Omega^{\mathrm{T}}}
\end{aligned}
$$

*Considering the worst case with $m_\Omega^{\mathrm{F}} = 1, m_v^{\mathrm{F}} = 0, v = 1, 2, \cdots, n$, then we can get $m_c^{\mathrm{Co}} = m_c^{\mathrm{T}}$.* $\square$

**Proposition 2:** *The $m_\Omega^{\mathrm{Co}}$ is monotonically increasing with $m_\Omega^{\mathrm{T}}$ and $m_\Omega^{\mathrm{F}}$.*

*Proof:*

$$
\begin{aligned}
m_\Omega^{\mathrm{Co}} &= \frac{m_\Omega^{\mathrm{T}} m_\Omega^{\mathrm{F}}}{\sum_{v=1}^{C} (m_v^{\mathrm{T}} m_v^{\mathrm{F}} + m_v^{\mathrm{T}} m_\Omega^{\mathrm{F}} + m_v^{\mathrm{F}} m_\Omega^{\mathrm{T}}) + m_\Omega^{\mathrm{T}} m_\Omega^{\mathrm{F}}} \\
&= \frac{1}{\sum_{v=1}^{C} \left( \frac{m_v^{\mathrm{T}} m_v^{\mathrm{F}}}{m_\Omega^{\mathrm{T}} m_\Omega^{\mathrm{F}}} + \frac{m_v^{\mathrm{T}}}{m_\Omega^{\mathrm{T}}} + \frac{m_v^{\mathrm{F}}}{m_\Omega^{\mathrm{F}}} \right) + 1}
\end{aligned}
$$

*As can be seen, $m_\Omega^{\mathrm{Co}}$ increases as $m_\Omega^{\mathrm{T}}$ and $m_\Omega^{\mathrm{F}}$ increase.* $\square$

### A.3 Description of datasets

**ECG** (Clifford et al., 2017) is from the 2017 PhysioNet Challenge focusing on classifying ECG recordings, where single-lead ECG measures four different underlying conditions of cardiac arrhythmias. **EMG** (Goldberger et al., 2000) consists of single-channel Electromyograms (EMGs) recorded from the tibialis anterior muscle of three healthy volunteers suffering from myopathy and neuropathy. **HAR** (Anguita et al., 2013) contains recordings of 30 health volunteers performing daily activities, including walking, walking upstairs, walking downstairs, sitting, standing, and lying. **Gesture** (Liu et al., 2009) contains accelerometer measurements of eight simple gestures that differ based on the paths of hand movement. **FD-A/FD-B** (Lessmeier et al., 2016) corresponds to

Faulty Detection Condition A (FD-A) and Faulty Detection Condition B (FD-B), which are generated by an electromechanical drive system that monitors the condition of rolling bearings and detects their failures. **SleepEEG** (Kemp et al., 2000) are collected from 82 healthy subjects and contains 153 whole-night sleeping electroencephalography (EEG) recordings. **Epilepsy** (Andrzejak et al., 2001) contains single-channel EEG measurements from 500 subjects and the corresponding brain activity is recorded for 23.6 seconds. The details about the UCR datasets can be found in (Chen et al., 2015).

## A.4 Description of baselines

**Baselines.** To answer Q1, we consider the following 10 time-series clustering methods in the comparison experiment.

- **DTCR** (Ma et al., 2019b): it integrates the temporal reconstruction and $k$-means objective into the seq2seq model. By proposing a fake-sample generation strategy and auxiliary classification task, it can learn cluster-specific temporal representations.

- **k-shape** (Paparrizos & Gravano, 2015): it relies on a scalable iterative refinement procedure and uses a normalized cross-correlation measure to consider the shapes of time-series.

- **SOM-VAE** (Fortuin et al., 2020): it overcomes the non-differentiability in discrete representation learning and presents a gradient-based version of the traditional self-organizing map (SOM) algorithm.

- **STCN** (Ma et al., 2022): it optimizes the feature extraction and clustering simultaneously, where an RNN conducts the reconstruction of time-series and a self-supervised module to obtain the clustering result.

- **TMEK** (Tang et al., 2021): it maps the raw time-series space into multiple kernel spaces via elastic distance measure functions, and resorts to the tensor-constraint-based self-representation subspace clustering approach.

- **TNC** (Tonekaboni et al., 2022): it uses the local smoothness of a signal's generative process to define neighborhoods in time-series. By using a de-biased contrastive objective, it learns time-series representations that are input to $k$-means to produce the clusteri.ng results.

- **TS3C$_{ch}$** (Guijo-Rubio et al., 2021): it consists of two stages, where a least squares polynomial technique is first used to segment the time-series and the hierarchical clustering is applied to all the mapped segmentations.

- **UMAP** (Péalat et al., 2023): it embeds the time-series onto higher-dimensional spaces and is in conjunction with HDBSCAN algorithm (Campello et al., 2013) to obtain the results.

- **USSL** (Zhang et al., 2019): it integrates the shapelet learning, shapelet regularization, spectral analysis and pseudo-label to automatically learn shapelets to group time-series samples.

- **VLSC** (Duan & Guo, 2023): it minimizes the inner time-series clustering error under time-series cover constraints, where the time-series lengths can be variable.

To answer Q3, we evaluate the embeddings learned by TIC on the other downstream tasks (i.e., classification and anomaly detection). The included baselines are 8 unsupervised representation learning methods for time-series.

- **activity2vec** (Aggarwal et al., 2019): it learns the representations with three components, the co-occurrence and magnitude of the activity levels in a time-series sample, neighboring context of the time-series, and promoting subject-invariance with adversarial training.

- **BTSF** (Yang & Hong, 2022): it utilizes the sample-level augmentation with a simple dropout on the time-series dataset and devise the iterative bilinear temporal-spectral fusion to encode the affinities of abundant time-frequency pairs.

- **CLOCS** (Kiyasseh et al., 2021): it encourages the time-series representations across space, time, and patients to be similar to one another for the physiological data.

- **MHCCL** (Meng et al., 2023): it exploits semantic information obtained from the hierarchical structure consisting of multiple latent partitions for multivariate time-series.

- **TFC** (Zhang et al., 2022): it learns the representations of time-series by positing that embedding a time-based neighborhood of a sample should be close to its frequency-based neighborhood.

- **Triplet** (Franceschi et al., 2019): it learns general-purpose representations by combining an encoder based on causal dilated convolutions with a novel triplet loss employing time-based negative sampling.

- **TS2Vec** (Yue et al., 2022): it performs contrastive learning in a hierarchical way over augmented context views and obtains a contextual representation for each timestamp.

- **TSTCC** (Eldele et al., 2022): it proposes time-series-specific weak and strong augmentations and learns discriminative representations in a contextual contrasting module.

### A.5 MORE STATISTICAL TEST

We also compare the performance of methods using the Friedman test and Nemenyi test by setting the significant level to 0.05, which is shown in the Fig.6. TIC significantly outperforms the baselines except for SOM-VAE and TS3C$_{ch}$ in all cases.

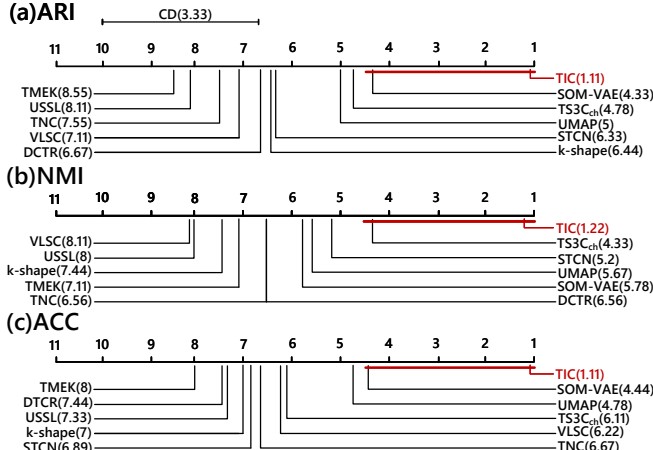

Figure 6: Comparison between TIC and other baselines with the Nemenyi test. The lowest (best) ranks are to the right, and thus methods on the right sides are considered to be better. The groups of baselines that are not significantly different from TIC and are connected by red.

### A.6 NMI, ACC AND RUNNING TIME RESULTS

Overall, our TIC model wins 235 and has tied performance on 34 out of 270 trials, when it is statistically compared with 10 baselines based on all the three metrics. Compared to the baselines, TIC consumes comparative runtime but achieves the best clustering accuracy.

### A.7 MORE ABLATION STUDY

We conduct other ablation studies to evaluate the importance of using the Dempster's rule in the developed TIC model. Concretely, we compare TIC model with the following variants:

- **TIC$_+$**: Dempster's rule is replaced by the addition $+$ to combine the mass functions $\mathbf{m}^{\text{Time}}$ and $\mathbf{m}^{\text{Freq}}$. For example, if $\mathbf{m}_i^{\text{Time}} = [0.8, 0.1, 0.1], \mathbf{m}_i^{\text{Freq}} = [0.7, 0.2, 0.1]$, the $\mathbf{m}_i^{\text{Comb}}$ is calculated as $m_{i1}^{\text{Comb}} = \frac{0.8+0.7}{0.8+0.7+0.1+0.2+0.1+0.1} = 0.75, m_{i2}^{\text{Comb}} = \frac{0.1+0.2}{2} = 0.15$ and $m_{i\Omega}^{\text{Comb}} = \frac{0.1+0.1}{2} = 0.1$;

- **TIC-cau/TIC-bol/TIC-tri**: Dempster's rule is replaced by other combination rule, i.e., the cautious conjunctive rule (Denœux, 2008), the bold conjunctive rule (Denœux, 2008), the parametric triangular-norm-based rule (Su et al., 2018), which do not rely on the assumption that the items of evidence are independent of each other;

Table 5: NMI, ACC (mean$_{\pm\text{std.deviation}}$) and running time of different algorithms on benchmark datasets. The ●/○ indicates whether TIC is statistically superior/inferior to a certain comparing baseline based on the paired $t$-test at a 0.05 significance level. The statistics of win/tie/loss are shown in the last row of each sub-table. The best and the second-best results, between which the performance gaps are shown in the row named "gap" in each sub-table, are colored blue and red.

| NMI | EMG | ECG | HAR | Gesture | FD-A | FD-B | SleepEEG | Epilepsy | UCR |
|---|---|---|---|---|---|---|---|---|---|
| DTCR | .802±.02● | .745±.01● | .653±.02● | .766±.02● | .862±.01● | **.822±.02○** | .492±.02● | .802±.02● | .663±.02● |
| $k$-shape | .752±.02● | .591±.01● | .745±.01● | .734±.02● | .836±.02● | .756±.03 | .692±.02● | .834±.03● | .843±.03● |
| SOM-VAE | .892±.03● | .632±.02● | .822±.02● | .888±.01 | .789±.03● | .682±.02● | .722±.02● | .929±.01● | .825±.02● |
| STCN | **1±0** | .745±.03● | .669±.02● | .793±.01● | .894±.02 | .726±.03● | .833±.02● | .802±.01● | .639±.02● |
| TMEK | .696±.02● | .653±.02● | .738±.01● | .859±.01● | .721±.02● | .730±.01● | .793±.03● | .952±.02 | .703±.01● |
| TNC | .743±.01● | .746±.02● | .654±.01● | .833±.02● | .751±.02● | .652±.01● | .799±.02● | .854±.01● | .738±.01● |
| TS3C$_{ch}$ | .752±.01● | .726±.01● | .753±.02● | .871±.02 | .754±.04● | .752±.03● | .921±.01 | .949±.01● | .874±.01 |
| UMAP | .863±.01● | .720±.01● | .610±.02● | .827±.02● | .714±.04● | .746±.03● | .840±.01● | .954±.01 | .882±.01 |
| USSL | .842±.01● | .702±.02● | .496±.01● | .568±.02● | .652±.02● | .705±.03● | .900±.02● | .895±.01● | .736±.02● |
| VLSC | **1±0** | .734±.01● | .578±.02● | .724±.02● | .630±.03● | .339±.02● | .754±.01● | .831±.02● | .731±.01● |
| TIC (ours) | **1±0** | **.786±.01** | **.887±.01** | **.898±.02** | **.900±.01** | **.784±.02** | **.942±.01** | **.965±.02** | **.891±.01** |
| gap | .108 | .040 | .065 | .010 | .006 | .038 | .021 | .011 | .009 |
| win/tie/loss | 8/2/0 | 10/0/0 | 10/0/0 | 8/2/0 | 9/1/0 | 8/1/1 | 9/1/0 | 7/3/0 | 8/2/0 |

| ACC | EMG | ECG | HAR | Gesture | FD-A | FD-B | SleepEEG | Epilepsy | UCR |
|---|---|---|---|---|---|---|---|---|---|
| DTCR | .792±.02● | .841±.01● | .621±.02● | .892±.03● | .906±.02● | **.938±.01** | .620±.01● | .921±.02● | .657±.02● |
| $k$-shape | .715±.02● | .669±.01● | .802±.03● | .856±.02● | .925±.01● | .837±.02● | .840±.01● | .931±.02● | .831±.01● |
| SOM-VAE | .885±.03● | .879±.02● | .839±.01● | .940±.01 | .862±.02● | .781±.02● | .874±.02● | .952±.02● | .841±.01● |
| STCN | **1±0** | .785±.02● | .805±.02● | .839±.01● | .919±.03● | .846±.02● | .826±.02● | .810±.01● | .605±.02● |
| TMEK | .742±.02● | .795±.02● | .800±.01● | .863±.01● | .752±.02● | .822±.01● | .793±.03● | .954±.02● | .678±.01● |
| TNC | .838±.01● | .921±.02 | .726±.01● | .896±.02● | .821±.03● | .675±.01● | .820±.03● | .829±.01● | .771±.01● |
| TS3C$_{ch}$ | .798±.02● | .842±.01● | .732±.01● | .889±.01● | .796±.03● | .726±.01● | .933±.01 | .950±.02● | .895±.02 |
| UMAP | .876±.01● | .846±.01● | .705±.02● | .909±.02● | .731±.02● | .863±.01● | .921±.01● | .988±.03 | .867±.01 |
| USSL | .932±.01● | .822±.01● | .734±.02● | .619±.02● | .658±.04● | .639±.03● | .885±.01● | .978±.01● | .751±.01● |
| VLSC | **1±0** | .846±.01● | .687±.02● | .879±.02● | .786±.03● | .602±.02● | .941±.01 | .977±.02 | .742±.01● |
| TIC (ours) | **1±0** | **.935±.01** | **.869±.01** | **.959±.01** | **.982±.01** | **.950±.02** | **.957±.01** | **.997±.01** | **.914±.02** |
| gap | .068 | .014 | .030 | .019 | .057 | .012 | .016 | .009 | .019 |
| win/tie/loss | 8/2/0 | 9/1/0 | 10/0/0 | 9/1/0 | 10/0/0 | 9/1/0 | 8/2/0 | 7/3/0 | 8/2/0 |

| Time | EMG | ECG | HAR | Gesture | FD-A | FD-B | SleepEEG | Epilepsy | UCR |
|---|---|---|---|---|---|---|---|---|---|
| DTCR | 1.65 | 412.2 | 113.2 | 8.5 | 213.2 | 324.3 | 3486.3 | 143.7 | 154.3 |
| $k$-shape | 0.32 | 541.1 | 132.7 | 14.3 | 300.9 | 452.9 | 3688.2 | 168.3 | 217.8 |
| SOM-VAE | 2.51 | 365.8 | 101.7 | 7.2 | 196.2 | 331.1 | 3348.2 | 121.9 | 169.8 |
| STCN | 0.85 | 5421.1 | 2117.3 | 3.4 | 1837.6 | 2913.2 | 27986.3 | 2684.9 | 186.9 |
| TMEK | 1.89 | 432.4 | 99.4 | 6.3 | 145.7 | 217.3 | 2987.6 | 119.3 | 197.9 |
| TNC | 2.07 | 500.7 | 145.3 | 5.2 | 164.8 | 213.7 | 3114.7 | 192.3 | 208.9 |
| TS3C$_{ch}$ | 1.54 | 472.1 | 123.6 | 10.9 | 151.8 | 207.9 | 3864.3 | 143.5 | 178.3 |
| UMAP | 1.25 | 625.1 | 200.6 | 7.6 | 275.9 | 423.9 | 3004.7 | 287.9 | 190.6 |
| USSL | 2.09 | 587.3 | 132.4 | 4.7 | 167.3 | 246.4 | 3654.8 | 169.3 | 183.7 |
| VLSC | 1.74 | 584.6 | 241.1 | 5.6 | 272.2 | 384.3 | 3845.1 | 300.8 | 223.7 |
| TIC (ours) | 2.09 | 576.3 | 135.3 | 6.8 | 206.2 | 312.9 | 3884.9 | 249.9 | 199.8 |

• TIC-comc/TIC-GC/TIC-yage/TIC-dubo/TIC-RCR: Dempster's rule is replaced by other combination rule, i.e., the COMC rule (Ma et al., 2019c), the GC rule (Du & Zhong, 2021), Yager's rule (Yager, 1987), Dubois-Prade's rule (Dubois & Prade, 1988), the RCR rule (Florea et al., 2009) which have their own ways to deal with the high-conflicted mass functions.

Table 6: ARI values (mean$_{\pm\text{std.deviation}}$) of different variants of TIC.

| ARI | EMG | ECG | HAR | Gesture | FD-A | FD-B | SleepEEG | Epilepsy | UCR | Average |
|---|---|---|---|---|---|---|---|---|---|---|
| TIC$_\times$ | .845 | .729 | .711 | .921 | .895 | .803 | .832 | .893 | .873 | |
| TIC$_+$ | .839 | .698 | .687 | .909 | .910 | .821 | .850 | .869 | .867 | |
| TIC-cau | .965 | .873 | .783 | .924 | .924 | .839 | .892 | .931 | .832 | |
| TIC-bol | .981 | .869 | .789 | .920 | .867 | .841 | .882 | .965 | .809 | |
| TIC-tri | **1** | .879 | .772 | .915 | .926 | .839 | .889 | .978 | .867 | |
| TIC-comc | .987 | .872 | .754 | .928 | .927 | .828 | .902 | .963 | .856 | |
| TIC-GC | .979 | .871 | .768 | .916 | .913 | .819 | .856 | .952 | .873 | |
| TIC-yage | **1** | .869 | .781 | .924 | .918 | .834 | .883 | .980 | .830 | |
| TIC-dubo | .981 | .853 | .789 | .916 | .904 | .842 | .879 | .980 | .815 | |
| TIC-RCR | **1** | .874 | .792 | .904 | .919 | .836 | .882 | .958 | .871 | |
| TIC (Full model) | **1** | .879 | .796 | .931 | .939 | .853 | .896 | .980 | .883 | |

Compared to the full model, the average ARI of TIC$_\times$ and TIC$_+$ decrease by $0.912 - 0.834 = 0.078$ and $0.912 - 0.833 = 0.079$. This suggests that it is more reasonable to use Dempster's rule to fuse the clustering results from time and frequency domains. The potential reasons for this result are twofold: (1) as shown in Eq.(9), Dempster's rule includes the $m_{ic}^{\text{Time}} \cdot m_{ic}^{\text{Freq}}$ (multiplication term) and $m_{ic}^{\text{Time}} \cdot m_{i\Omega}^{\text{Freq}} + m_{i\Omega}^{\text{Time}} \cdot m_{ic}^{\text{Freq}}$ (addition term); (2) Dempster's rule considers additionally the uncertainty $m_{i\Omega}^{\text{Time}}$ and $m_{i\Omega}^{\text{Freq}}$ (in term $m_{i\Omega}^{\text{Time}} \cdot m_{ic}^{\text{Freq}} + m_{ic}^{\text{Time}} \cdot m_{i\Omega}^{\text{Freq}}$), when calculating $m_{ic}^{\text{Comb}}$. To further illustrate the contribution of combining the results from time and frequency domains, we show the cosine distance between $\mathbf{g}_i^{\text{time}}$ and $\mathbf{g}_i^{\text{Freq}}$ of the same sample in Fig.4(a). "W/o Comb" means that the cosine distance is calculated between the frequency embeddings learned from "W/o

$\mathcal{L}^{\text{Time}}$" and the time embeddings learned from "W/o $\mathcal{L}^{\text{Freq}}$". One can find that the cosine distance is smaller by combining the results from time and frequency domains. It means that combining the results from these two domains indeed enforces the time- and frequency-based embeddings closer to each other. Taking the FD-A dataset as an example, the average cosine distance decreases from 1.924 to 1.557, i.e., 19.1% by considering the combination.

Compared with TIC-cau and TIC-bol, TIC performs better on all the benchmark datasets. The reasons are: the cautious conjunctive rule uses the intersection operation and loses useful information; the bold conjunctive rule uses the union operation and takes into account confusing information when making a decision. Comparing TIC with TIC-tri, TIC-tri only has the same ARI as TIC in 2 cases but has lower ARI on the other 7 cases. It is because that the parametric triangular-norm-based rule is more sensitive to the hyper-parameters, and reasonable hyper-parameter values are difficult to choose in both of the time and frequency domains. This also suggests that the mass functions associated with the clustering results from the time and frequency domains are independent of each other in the time series clustering problem studied in our paper. Besides, only TIC-comc in the 5 variants aiming to tackle high-conflict have higher ARI than TIC on SleepEEG dataset, because the conflict values between the results of time and frequency domains are small.

Other 5 variants

- TIC-SVM, TIC-LR, TIC-FC and TIC-LSTM: the outputs from time- and frequency- domains are treated as features to train SVM, Logistic Regression, fully connected layers, and LSTM;

- TIC-max: the maximum probability given in networks from time- and frequency- domains is chosen as the final output probability.

are considered to show that using evidence theory to combine the results from time and frequency domains are superior to other fusion methods. As shown in Table 7, Using evidence theory has the best ARI in 7 of 9 cases, showing that fusing the results via evidence theory is better than other methods.

Table 7: ARI values (mean$_{\pm\text{std.deviation}}$) of different variants of TIC.

| ARI | EMG | ECG | HAR | Gesture | FD-A | FD-B | SleepEEG | Epilepsy | UCR | Average |
|---|---|---|---|---|---|---|---|---|---|---|
| TIC-SVM | .987 | .875 | .779 | .930 | .928 | .842 | .882 | .962 | .869 | |
| TIC-LR | **1** | .839 | .782 | .929 | .928 | .844 | .885 | .974 | .870 | |
| TIC-FC | **1** | .860 | .769 | .934 | .932 | **.859** | .876 | **.980** | .863 | |
| TIC-LSTM | .993 | .853 | .739 | **.945** | .917 | .851 | .879 | .974 | .876 | |
| TIC-max | .976 | .842 | .754 | .928 | .921 | .809 | .882 | .956 | .879 | |
| TIC (Full model) | **1** | **.879** | **.796** | .931 | **.939** | .853 | **.896** | **.980** | **.883** | |

## A.8 EMBEDDING EVALUATION: TIME-SERIES ANOMALY DETECTION.

We evaluate how TIC performs on a sample-level anomaly detection task, which aims to detect abnormal time-series samples. We build two subsets of FD-B and ECG datasets. The former contains 1000 samples where 900 undamaged bearings are considered "normal" and 100 damaged samples are "outliers"; while the latter has 2000 samples where 1800 normal sinus rhythm recordings are considered "normal" and 200 atrial fibrillation recordings are "outliers". We randomly select half of the "normal" samples as the training data, of which the embeddings learned by different models are used to train the one-class SVM. The time- and frequency-based embeddings learned by TIC are also concatenated $[\mathbf{g}^{\text{Time}}; \mathbf{g}^{\text{Freq}}]$. In Table 8, we report the performance on the anomaly detection task in terms of Precision, Recall, F1-score and AUROC. TIC achieves the best performance in 6 out of 8 cases. On the ECG dataset, TIC outperforms the second-best model (i.e., activity2vec) by a margin of 3.3% in AUROC. It shows that TIC can effectively detect the abnormal samples in mechanical devices and ECGs.

## A.9 EFFECT OF AUGMENTATION

As shown in Section 3, data augmentations are adopted in both the time- and frequency-based contrastive modules. We explore the effects of augmentation techniques in these two modules separately. In each trial, the corresponding setting is changed while the other ones are the same as the default settings.

Table 8: Performance on time-series anomaly detection. The subsets of FD-B (1000 samples) and ECG (2000 samples) are used. These two subsets are highly imbalanced (90% normal samples and 10% abnormal samples).

| FD-B | Precision | Recall | F1-score | AUROC |
|---|---|---|---|---|
| activity2vec | $.8054_{\pm.034}$ | $.7145_{\pm.037}$ | $.7432_{\pm.023}$ | $.7911_{\pm.029}$ |
| BTSF | $.6854_{\pm.015}$ | $.6274_{\pm.017}$ | $.6653_{\pm.021}$ | $.6741_{\pm.014}$ |
| CLOCS | $.8412_{\pm.025}$ | $.7465_{\pm.028}$ | $.8222_{\pm.019}$ | $.8126_{\pm.021}$ |
| MHCCL | $.6210_{\pm.043}$ | $.5745_{\pm.036}$ | $.6009_{\pm.040}$ | $.6123_{\pm.023}$ |
| TFC | $\mathbf{.8547}_{\pm.037}$ | $\mathbf{.7698}_{\pm.029}$ | $\mathbf{.8274}_{\pm.019}$ | $\mathbf{.8602}_{\pm.033}$ |
| Triplet | $.7321_{\pm.034}$ | $.6547_{\pm.045}$ | $.6987_{\pm.043}$ | $.7201_{\pm.031}$ |
| TS2Vec | $.6813_{\pm.022}$ | $.6134_{\pm.027}$ | $.6423_{\pm.019}$ | $.6731_{\pm.015}$ |
| TSTCC | $.6354_{\pm.019}$ | $.4517_{\pm.066}$ | $.4968_{\pm.056}$ | $.8022_{\pm.044}$ |
| TIC (ours) | $\underline{\mathbf{.8641}}_{\pm.025}$ | $\underline{\mathbf{.7652}}_{\pm.013}$ | $\underline{\mathbf{.8359}}_{\pm.024}$ | $\underline{\mathbf{.8563}}_{\pm.033}$ |
| ECG | Precision | Recall | F1-score | AUROC |
| activity2vec | $.7584_{\pm.011}$ | $.6124_{\pm.036}$ | $.7022_{\pm.073}$ | $\mathbf{.7598}_{\pm.026}$ |
| BTSF | $\mathbf{.7752}_{\pm.027}$ | $\mathbf{.7124}_{\pm.034}$ | $.7413_{\pm.029}$ | $.7581_{\pm.029}$ |
| CLOCS | $.4861_{\pm.049}$ | $.4035_{\pm.056}$ | $.4491_{\pm.038}$ | $.4727_{\pm.023}$ |
| MHCCL | $.5689_{\pm.015}$ | $.4875_{\pm.019}$ | $.5471_{\pm.024}$ | $.5563_{\pm.023}$ |
| TFC | $.7654_{\pm.037}$ | $.7035_{\pm.029}$ | $\mathbf{.7503}_{\pm.021}$ | $.7429_{\pm.033}$ |
| Triplet | $.5789_{\pm.027}$ | $.5067_{\pm.023}$ | $.5417_{\pm.028}$ | $.5561_{\pm.019}$ |
| TS2Vec | $.6857_{\pm.015}$ | $.6138_{\pm.019}$ | $.6587_{\pm.021}$ | $.6701_{\pm.020}$ |
| TSTCC | $.7345_{\pm.031}$ | $.6538_{\pm.024}$ | $.6993_{\pm.016}$ | $.7235_{\pm.019}$ |
| TIC (ours) | $\underline{\mathbf{.7958}}_{\pm.021}$ | $\underline{\mathbf{.7216}}_{\pm.024}$ | $\underline{\mathbf{.7645}}_{\pm.026}$ | $\underline{\mathbf{.7856}}_{\pm.019}$ |

In the time-based contrastive module, we fix the augmentation as jittering, scaling and time-shift for all samples, instead of randomly choosing one augmentation from the augmentation bank $\mathcal{B}^{\text{Time}}$ (consisting of these three augmentations) for each sample. The comparison result is shown in the rightmost sub-table of Table 9. As can be seen, random selection covers diverse augmentations that allow the encoder $H_T(\cdot)$ to learn better and more robust embedding $\mathbf{g}^{\text{Time}}$, which improves the clustering performance.

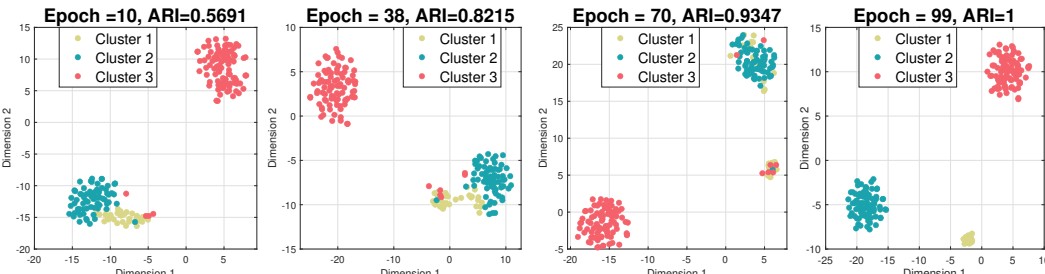

Figure 7: T-SNE visualization of the concatenated embedding $[\mathbf{g}_i^{\text{Time}}; \mathbf{g}_i^{\text{Freq}}]$ for EMG dataset in different epochs. As the learning proceeds, the embeddings from the same cluster are gradually grouped together.

We consider three types of change in frequency-based augmentations. (1) We explore the number of components manipulated, where the results are shown by the sub-table titled by *Components* in Table 9. One can find that the performance of TIC model tends to deteriorate when perturbing multiple frequency components. This degradation is attributed to the substantial changes in the time-domain of augmented samples. Consequently, these augmented samples become readily distinguishable by a contrastive module, resulting in suboptimal contrastive encoders. (2) We then explore the adjustment size of amplitude in sub-table *Amplitude* of Table 9. As can be seen, manipulating the amplitude does not significantly affect the performance of TIC model. (3) In sub-table *Bands* of Table 9, we explore the band that the manipulated component belongs to. The Low-/high-frequency band corresponds to the first/second half of the frequency spectrum, and contributes to slow/fast variations in the time domain. In a physiological time-series dataset (e.g., SleepEEG), the low band contains most information and manipulating a low-frequency component leads to higher ARI. On the contrary, high-band components are more informative than low-band ones in a mechanical time-series dataset (e.g., FD-A). Thus, perturbing high-band components outperform low-band augmentations.

## A.10 VISUALIZATION OF CLUSTERING RESULTS

Figure 7 shows the t-SNE Van der Maaten & Hinton (2008) plot of the embeddings for EMG dataset learned by TIC model. For each sample, we concatenate the embeddings $[\mathbf{g}_i^{\text{Time}}; \mathbf{g}_i^{\text{Freq}}]$. As the

Table 9: ARI with different augmentation techniques. Terms *Components*, *Amplitudes* and *Bands* refer to the frequency-based augmentations. *Components* means how many components are manipulated, *Amplitude* means the adjustment size of amplitude and *Bands* means the perturbation is performed on a low- or a high-frequency component. *Time domain* means the augmentation $\widetilde{\mathbf{x}}_i^{\text{Time}}$ are randomly selected from the bank $\mathcal{B}_i^{\text{Time}}$ {Jittering, Scaling Shift} or a fixed augmentation technique.

| ARI | Components | | | Amplitude | | | | Bands | | Time domain | | | |
|---|---|---|---|---|---|---|---|---|---|---|---|---|---|
| | $\beta = 1$ | $\beta = 3$ | $\beta = 5$ | $\gamma = 0.1$ | $\gamma = 0.5$ | $\gamma = 0.9$ | $\gamma = 1.1$ | Low | High | Random | Jittering | Scaling | Shift |
| SleepEEG | **0.8961** | 0.8751 | 0.8245 | 0.8952 | **0.8961** | 0.8934 | 0.8942 | **0.8994** | 0.8802 | **0.8961** | 0.8726 | 0.8542 | 0.8793 |
| FD-A | **0.9392** | 0.9157 | 0.9013 | 0.9406 | 0.9392 | **0.9410** | 0.9375 | 0.9321 | **0.9457** | **0.9392** | 0.9103 | 0.9261 | 0.9245 |

learning proceeds, the embeddings learned by TIC from the same cluster are gradually grouped together. In particular, samples clearly form 3 clusters at epoch $= 99$, corresponding to the ARI=1 of TIC shown in Table 5. The visualization results further demonstrate the well embedding ability of our model.

