# OpenReview forum: "Trusted and Interactive Clustering for Time-Series Data"
_ICLR.cc/2025/Conference — ICLR 2025 Conference Withdrawn Submission_

### Official Review · Reviewer_UJk1 · 2024-10-16

**Soundness:** 3
**Presentation:** 2
**Contribution:** 2
**Rating:** 3
**Confidence:** 3

**Summary:**

The paper proposes an approach to cluster time series using a contrastive learning approach to learn meaningful representations.
To implement contrastive learning, time series augmentations based both on the time and frequency domain are used.
Rather than directly generating cluster assignments, the proposed model outputs parameters of a Dirichlet distribution. The latter are further processed to obtain the final cluster assignments.
A k-means algorithm is run from time to time during training to generate cluster assignments that guide the optimization.

**Strengths:**

The idea of inserting contrastive learning within a probabilistic framework is interesting in principle. However, the proposed solution and implementation seem flawed.

**Weaknesses:**

The most severe of my concerns is that one of the main contribution of the paper, which is the contrastive learning based both on time and frequency agumentation is not a novel contribution but is proposed in

Zhang, Xiang, et al. "Self-supervised contrastive pre-training for time series via time-frequency consistency." Advances in Neural Information Processing Systems 35 (2022): 3988-4003.

The approach is basically the same and even some considerations/discussions are taken from that paper. Despite the paper from Zhang et al. is cited, the authors do not say that contrastive learning approach based on time and frequency augmentation comes from there. On the other hand, the authors make it look like is their own contribution.

My second concern is about the clarity of the presentation: the paper is rather confusing and difficult to follow. There are a lot of different concepts mixed together, often in a non-organic manner, and none of them seem particularly novel or well justified.

The whole part about the probabilistic framework, uncertainty quantification, and trustworthiness seems badly implemented. the authors say that their decoder produces parameters for a Dirichelet distribution. However, there are neither sampling procedures nor variational losses in the framework that justify the use of the probabilistic framework. It seems to me that the authors try to address this by introducing concepts such as "Subjective logic" and "Dempster’s rule", which I found rather confusing. In fact, I don't understand how much different it would be in practice to produce the cluster assignments directly from the FC decoder.

Finally, running a k-means while training a deep learning framework seems very unpractical and costly to implement, both in terms of time and space complexity. The fact that one should resort to such a solution hints that the proposed framework is unstable and unable to learn good cluster assignments.

**Questions:**

- Why the work of Zhang et al. was not properly described, saying that they introduced a contrastive learning framework that ensures consistency between time and frequency augmentations?
- What performance would you get by producing directly the cluster assignments m_c from the FC decoder?

**Details Of Ethics Concerns:**

One of the main contributions of the paper is actually proposed in the work

Zhang, Xiang, et al. "Self-supervised contrastive pre-training for time series via time-frequency consistency." Advances in Neural Information Processing Systems 35 (2022): 3988-4003.

The authors are not transparent about it and sell the same idea as their own contribution.

---

> ### Author Response · Authors · 2024-11-19
> **Q1: About the reference of paper [1]**
>
> The fact that paper [1] also employs a contrastive learning technique combining the time- and frequency- domain lead to your misunderstanding that we should place more emphasis on citing [1]. This also have influenced reviewer wHWj's comments. We would like to address this matter in our urgent response.
>
> ### **A1.1: About contribution**
>
> Our paper makes two key different contributions:
>
> 1. Quantifying the uncertainty of clustering results from both the time and frequency domains (Contribution 2).
> 2. Proposing an interactive clustering framework (Contribution 3).
>
> These contributions are entirely unrelated to [1]. Reviewer nLrp recognized these as **''...novel directions...''**.
>
> ### **A1.2: We treat [1] as a baseline**
>
> We included the method from [1], TFC, as a baseline for comparison (see Table 4 and Table 8). Our method significantly outperformed TFC across benchmark datasets on two common tasks—classification and anomaly detection.
>
> ### **A1.3: Significant technical differences**
>
> Our similarity with [1] lies in the use of contrastive learning from both the time and frequency domains. However, the essence of our work is fundamentally different. To clarify, we will explain this distinction through the construction of positive and negative pairs.
>
> Let $z_i^{time}$, $z_i^{freq}$, $\tilde{z}_i^{time}$, and $\tilde{z}_i^{freq}$ denote the embeddings (from time and frequency domains) of a sample $x_i^{time}$ and its augmentation $\tilde{x}_i^{time}$.
>
> - **Positive pairs in [1]**:
>   $(z_i^{time}, \tilde{z}_i^{time})$, $(z_i^{freq}, \tilde{z}_i^{freq})$, $\boldsymbol{(z_i^{time}, z_i^{freq})}$, $\boldsymbol{(\tilde{z}_i^{time}, \tilde{z}_i^{freq})}$
> - **Negative pairs in [1]**:
>   $(z_i^{time}, z_j^{time})$, $(\tilde{z}_i^{time}, \tilde{z}_j^{time})$, $(z_i^{freq}, z_j^{freq})$, $(\tilde{z}_i^{freq}, \tilde{z}_j^{freq})$
> - **Positive pairs in our paper**:
>   $(z_i^{time}, \tilde{z}_i^{time})$, $(z_i^{freq}, \tilde{z}_i^{freq})$
> - **Negative pairs in our paper**:
>   $(z_i^{time}, z_j^{time})$, $(\tilde{z}_i^{time}, \tilde{z}_j^{time})$,  $(z_i^{freq}, z_j^{freq})$, $(\tilde{z}_i^{freq}, \tilde{z}_j^{freq})$, $\boldsymbol{(z_i^{freq}, \tilde{z}_j^{freq})}$, $\boldsymbol{(z_i^{time}, \tilde{z}_j^{time})}$
>
> The bold part highlight where our approach diverges from [1], where [1] includes two additional positive pairs, and our paper includes two additional negative pairs.
>
> - In [1], additional positive pairs are constructed to enforce the model to **learn better embeddings based on the idea of ''time-frequency consistency''**. In this way, they fuse the information from time and frequency domains.
> - We fuse the results of FC layers from time and frequency domains, and **achieve better embeddings by ''enabling interactive learning between the fused clustering results and the pseudo-labels generated by k-means''**. This is fundamentally different from the core idea of [1].
> - We construct more negative pairs to enhance the separation between clusters, thereby achieving more distinguishable clustering results.
>
> ### **A1.4: Different focus**
>
> The focus of our work is entirely different from [1]. While [1] studies pre-training for time-series models, we address time-series clustering.
>
> ### **A1.5: Combining time- and frequency-domain contrastive learning is a very basic technique used in earlier papers.**
>
> It could be illogical to claim that our work originates from [1].
>
> - The use of data augmentation techniques in both time and frequency domains has been well-established, e.g., [2-3].
> - Combining time and frequency domains in representation learning existed prior to [1], as demonstrated by works in [4-6]. Does this mean that [1]'s method is not original?
>
> In [1], the authors claimed to be the first to apply frequency-domain data augmentation techniques in contrastive learning for solving the **pre-training** problem of time-series models. Similarly, we clearly stated in Contribution 1 that we are the first to apply frequency-domain data augmentation techniques to time-series **clustering**.
>
> **Reference**
>
> [1] Zhang X, Zhao Z, Tsiligkaridis T, et al. Self-supervised contrastive pre-training for time series via time-frequency consistency. *Neurips*, 2022.
>
> [2] Cui X, Goel V, Kingsbury B, et al. Data augmentation for deep convolutional neural network acoustic modeling. *ICASSP*, 2015.
>
> [3] Wen Q, Sun L, Yang F, et al. Time series data augmentation for deep learning: A survey. * arXiv:2002.12478*, 2020.
>
> [4] Liu D, Wang T, Liu S, et al. Contrastive self-supervised representation learning for sensing signals from the time-frequency perspective. *ICCCN*, 2021.
>
> [5] Woo G, Liu C, Sahoo D, et al. CoST: Contrastive learning of disentangled seasonal-trend representations for time series forecasting. *ICLR*, 2022.
>
> [6] Yang L, Hong S, et al. Unsupervised time-series representation learning with iterative bilinear temporal-spectral fusion. *ICML*, 2022.

---

> ### Author Response · Authors · 2024-11-19
> **Q2: About the writing of our paper**
>
> It may be subjective to say our paper is "rather confusing and difficult to follow.
>
> ### **A2.1: About the "Uncertainty Quantification" section.**
>
> We kindly request that you carefully review the **Uncertainty Quantification** section (lines 198–208, lines 216–232) again. In summary, we utilize **Subjective Logic** [1] to associate the parameters of the Dirichlet distribution, $\alpha_c, c=1,2,\cdots,C$, with the output of the FC layer, $\textbf{p}_{\rm{T}}$
>
> $=[p_{1}^{\rm{T}}, p_{2}^{\rm{T}}, \cdots, p_{C}^{\rm{T}}]$
>
> Then, the parameter $\alpha_c$ is calculated as:
>
> $$
> \alpha_c = p_c + 1.
> $$
>
> The mass function $\textbf{m} = [m_1, m_2, \cdots, m_C, m_{\Omega}]$ is determined as follows:
>
> $$ \begin{cases} m_c = \frac{p_c}{S}=\frac{\alpha_c-1}{S} & c=1,2,\cdots,C \\\\ m_{\Omega}=\frac{C}{S} &
> \end{cases} $$ where $S=\sum_{c=1}^C (p_c+1)=\sum_{c=1}^C \alpha_c$ is the Dirichlet strength [1]
>
> Through these calculations, the clustering results of a sample **$x_i$** in the time domain or frequency domain can be represented in the form of a mass function. As explained in the Preliminaries section of our paper, in evidence theory, **$m_{\Omega}$ represents the measure of uncertainty in the clustering results of $x_i$ in the time or frequency domain**.
>
> Using **Dempster's Rule** (Eq.(2) in Preliminaries section), the clustering results from the time domain and frequency domain can be fused to obtain a reliable clustering result. This fusion process is described in detail in the **Combining the Clustering Results** section.
>
> Only through the above process, we can **quantitatively describe the uncertainty** in the clustering results of each sample from the time and frequency domains and **then fuse the results**. Directly using the output from the FC decoder fail to describe the uncertainty and cannot be directly fused.
>
>
>
> ### **A2.2: About the outcome directly using the FC's output.**
>
> We kindly ask that you pay closer attention to our ablation experiments, shown in Table 3, Table 6, and Table 7. We discuss the results in two cases:
>
> -We still use the results from both the time and frequency domains without uncertainty quantification. This means that the fusion method does not use evidence theory but instead applies direct addition, multiplication, or selects the maximum probability given by the networks from the time- and frequency-domain results as the final output probability. These 3 variants are denoted as TIC+, TICx, and TIC-max. We present the experimental results for these three variants in Table 6 and Table 7, where our method (TIC-full) shows a significant performance improvement over these variants.
>
> -We only use the result from either the time or the frequency domain, without fusion. We provide a comparison between our method and these two variants in Table 3, where our method still achieves better performance.
>
> ### **A2.3: About the concepts and notions in our paper.**
>
> We believe you may need to read our paper more carefully, because Reviewers wHWj and nLrp have both acknowledged the quality of our writing.
>
> - **Reviewer wHWj :** “Additionally, the method consists of various moving parts which makes it challenging to describe each component in a comprehensive manner. However, the authors did a great job in doing so. While the notation feels overloaded at times, the overall flow is quite good. Provided figures support this flow nicely.”
> - **Reviewer nLrp:** “Architecture diagram (Figure 1): Well annotated...,” “Preliminaries: Written with great clarity...,” “Methodology: Existing concepts, definitions, and theorems used are well-referenced...,” “Notations and Problem Statement are very clear...” and so on.
>
>
> **Reference**
>
> [1] Audun Jsang. Subjective Logic: A formalism for reasoning under uncertainty. Springer Publishing Company, Incorporated, 2018.

---

> ### Author Response · Authors · 2024-11-19
> **Q3: About the complexity of running k-means**
>
> ### **A3.1: K-means is a particularly efficient algorithm.**
>
> In our paper, the total complexity of k-means is approximately $O(I \times N \times K \times L)$, where $N$ is the number of samples, $L$ is the length of the time series, $I$ is the number of iterations, and $K$ is the number of clusters.
>
> The complexity of training our model (which includes two transformers and two 3-layer fully connected modules) is $O(N \times L \times (L \times 128 \times K)^2)$, where $K$ is the number of clusters and the term $(L \times 128 \times K)$ represents the number of neurons. The complexity of k-means is much smaller than that of the training models.
>
>  ### **A3.2: We have made specific designs for runtime efficiency.**
>
> As mentioned in the **Interactive Learning Module** section (line 272), we stated: "Run k-means every $t$ epochs." In the experiments section (line 387), $t$ is set to 20.
>
> This design was also appreciated by Reviewer nLrp: "Figure 5 very well demonstrates the need to avoid frequent updates using k-Means..."
>
>  ### **A3.3: We provide the proportion of runtime that k-means occupies throughout the entire training process.**
>
> || EMG | ECG | HAR | Gesture | FD-A | FD-B | SleepEEG | Epilepsy | UCR |
> |----------|----------|----------|----------|----------|----------|----------|----------|----------|----------|
> | Proportion | 0.12%    | 0.23%    | 0.45%    | 0.09%    | 0.34%    | 0.18%    | 0.21%    | 0.07%    | 0.65%    |
>
> It can be observed that the runtime of k-means occupies a very small proportion across all benchmark datasets.

---

> ### Comment · Reviewer_UJk1 · 2024-11-22
>
> Thanks for your detailed answers.
>
> I admit that I missed some of the results presented in the experimental section and that I might have not understood some parts of the uncertainty quantification section. As such, I will slightly raise my score and lower my confidence.
>
> With that said, I still find that the method is rather (perhaps unnecessary) complex and difficult to follow, especially for a reader like me who is not familiar with Subjective Logic, Dempster's rule, etc...
>
> Most importantly, the contrastive learning part based on temporal and frequency components is not your contribution, while the paper clearly suggests otherwise. I find this is not honest and, as such, I am not comfortable with accepting this work at this time. The paper should be rewritten to make clear what are the novelties and the contributions w.r.t. the existing body of work. For example, you should have included the discussion you provided in your first answer, which shows that the novelty is much more incremental compared to proposing from scratch the time/frequency contrastive learning.

---

### Official Review · Reviewer_nLrp · 2024-10-23

**Soundness:** 3
**Presentation:** 2
**Contribution:** 3
**Rating:** 6
**Confidence:** 4

**Summary:**

The model named TIC (Trusted and Interactive Clustering) tries to develop a methodology for clustering by fusing information from two distinct views: time and frequency domain. With strong backing from evidence theory and mass functions, the method learns robust embeddings from each domain using the respective contrastive loss. The learned embeddings are combined and an additional interactive loss using K-Means is additionally used to improve the clustering performance.

**Strengths:**

Introduction :
The second and third motivations of dynamically weighing the information from multiple views and combining them respectively for improving clustering results are novel directions.
Challenges associated with each motivation and the contributions from the work are clearly described.
Architecture diagram (Figure 1):
Well annotated and gives the overall idea of the methodology
Preliminaries :
Written with great clarity such that it is comprehensible to those who are not aware of evidence theory and mass function;
Focusing well on how they are fine-tuned towards the clustering and integration objectives.
Methodology:
Existing concepts, definitions, and theorems used are well-referenced, except in the "interactive learning module"
Notations and Problem Statement are very clear
The motivation for the design of the augmentation bank is well-written
Uncertainty quantification is well established with strong theoretical background
It is theoretically proven why the combined results are "trusted"
Figure 3 beautifully illustrates the "class collision" issue
Experiments :
Experimental evaluation on a large number of datasets against a variety of baselines for tasks of clustering and above (classification & anomaly detection) proves the effectiveness of the method
Statistical significance tests are used to prove that the proposed method is statistically different from other baselines
Excellent Ablation studies prove the significance of each loss component towards the model performance; A very detailed ablation study in the appendix for analyzing the contribution of each component further instantiates that the each of them is critical towards the model development
Figure 4b clearly illustrates how interactive loss contributes to better clustering
Figure 5 very well demonstrates the need to avoid frequent updates using k-Means and how the best frequency is chosen; The Subsequent explanation in "Hyperparameter sensitivity of t" is also well-written
Appendix :
A thorough literature study has been done as evident from Appendix A.1
Proofs for the propositions further confirm the "trusted" claim of the method
Baselines and datasets are well-described
Showing the effect of various perturbations in Table 9 conveys the idea of choosing the necessary techniques to alter the signals

**Weaknesses:**

Introduction :
It is mentioned that "there have been few clustering works which already attempted fusing time and frequency information"; Hence I feel it is not appropriate to give the first motivation for this work as "incorporating frequency information to enhance the ability to detect clusters."; It would have been more appropriate if it was mentioned that attempting this direction motivates other goals of weighing the contributions from each domain and incorporating them for clustering, which are novel.
Architecture diagram :
I guess the update decision function from k-Means indicates the epoch at which pseudo-labels are regenerated. If so, a better representation of the same is desirable.
Preliminaries :
Is the variable "m" missing from the definition "A mass function, .. is defined as "?
What is meant by "vacuous" in the phrase "vacuous mass function"?
Methodology :
It can be the case that the perturbations from the augmentation bank could be minor. Then the perturbed signals will not be contrastive enough and their cosine similarity could be high. Hence the claim that the augmentations used in both the modules are "contrastive" is weak, unless there is a strong reasoning. A more detailed explanation is desirable on how the perturbations are done: whether it is the same for the entire dataset; during training or random throughout
Contrastive loss functions (Equations 3,4):
A major basis for the work is that two distinct views for the time and frequency domain are formed. Hence is it justified to use the same value for the hyperparameter "τ" in both cases?
In the numerator it is shown that the similarity function is applied to embeddings of positive pairs; whereas the denominator uses "HF (x_j)" instead of the corresponding embedding g_j while applying the similarity function to negative pairs. What is the rationale behind it given that both are equivalent?
What is the reasoning behind using indicator functions in the denominator to exclude similarities between positive pairs? Because the loss function is in similar lines that of sigmoid and also conventional normalization techniques where such an indicator function is not used. Is it the same as the "class collision issue" mentioned in Remark 2? Still, it is not intuitive why the indicator function in contrastive loss can avoid class collision in interactive loss.
The concept of pseudo-labels is not properly explained or referenced; without which the role of K-Means looks incomplete
Equation 11 of the Interactive learning module is under-referenced where techniques like digamma function, and Dirichlet strength are used; Also what is the background with which conventional cross-entropy is modified? Theoretical backing for deriving equations 12,13 seem missing
Though Figure 3 gives a nice illustration of the "class collision issue"; I feel the term is misleading; It is mentioned that the issue represents the scenario where all clusters consist of only one sample; How does it convey the term "collision"?
The claim that "our TIC method can also accommodate multi-variate time-series" is strong and important in time-series analysis. However, it is not mentioned how the method presently done for univariate data can be easily extended to the multi-variate case.
Experiments :
"Implementation Details" :
 it is written that the "softmax layer is replaced with the RELU to ensure that the network outputs are non-negative"; But is the statement valid, because the output of softmax is also strictly non-negative?
 It is mentioned that "2-norm penalty coefficient of 0.0005" is used. However, there is no mention of the 2-norm regularization being used in the methodology and the purpose it serves.
"Ablation Study" :
It is valid that removing the interactive loss component leads to performance degradation due to "losing the basic clustering information from K-Means". But it is not clear why it gives the worst performance. The reasoning for the connection between "class collision issue" and "positive pairs" is still vague.
Figure 4 Illustration :
I feel there is a typo as one of the embedding notations should include "Freq" instead of "Time".
It is true that the time and frequency embeddings of the same sample should come close in the embedding space, But it is not clear how the model affects the cosine distance with and without their combination in the illustration though the description was later found in the appendix. It would be appropriate if the appendix description is included in the figure illustration itself.
"Hyperparameter sensitivity": Usually sensitivity experiments are used to demonstrate the model's robustness in the context of varying values of a critical hyperparameter. Here there are many hypermeters used in the model out of which only "t" is chosen; Also the demonstration is regarding how the best value is chosen and not on model robustness;
I think there is a typo in "Notions and problem formulation". Is it "Notations and .."?
Appendix :
Time Series Anomaly Detection baselines are not recent, state-of-the-art such as MTGFlow, TranAD, and GDN and are not evaluated using benchmark datasets such as SWaT, WADI, SMAP; So the results, though favorable to the proposed method cannot be considered as a breakthrough for this downstream task.
Figure 7 shows that the embeddings form distinct clusters towards the last epoch. But a strange observation is that the cluster distinction is not apparent till epoch 70 i.e. the first 2 plots essentially show the samples being distributed among two clusters; 3rd plot (epoch 70) shows 3 clusters and by the last plot (epoch 99) they become well distinct. Is there any justification for the same?
What is the unit of measure of the running time reported in Table 5?

**Questions:**

As above

---

### Official Review · Reviewer_wHWj · 2024-10-29

**Soundness:** 3
**Presentation:** 3
**Contribution:** 2
**Rating:** 3
**Confidence:** 4

**Summary:**

The authors propose a novel time series clustering method rooted in evidence theory. They approach relies on two embeddings, one in the time and one in the frequency domain. The resulting embeddings are used to parameterize a Dirichlet distribution which is then used to compute a mass function over the inputs (once for the time and once for the frequency domain). The mass functions are combined in an interactive loss which considers pseudo labels from a k-means clustering on the time series embeddings making sure to align the D. distribution with the k-means results. The authors provide evidence that their method results in superior clustering performance and good performance in secondary tasks such as anomaly detection and time series classification. Lastly, an ablation study is performed which sheds light on the importance of the involved loss terms.

**Strengths:**

The authors provide an exhaustive study and evaluation of the proposed method. They compare the method to various competing methods and dive deep into the mechanisms of their approach (ablation study). Additionally, the method consists of various moving parts which makes it challenging to describe each component in a comprehensive manner. However, the authors did a great job in doing so. While the notation feel overloaded at times, the overall flow is quite good. Provided figures support this flow nicely.

**Weaknesses:**

Statistical significance: The authors indicate statistical significance in Table 2. However, it seems they are not correcting for multiple hypotheses. Given the number of comparison methods and data set, I'd recommend adding a Critical Difference Plot[1] to underline statistical significance.
Algorithm efficiency: It is unclear how efficient the algorithm is. It would be great to see for each data set how long training takes and also how long training/inference of the competing methods take (e.g., adding a respective table).
Preliminaries: The background on Evidence Theory seems rather short. In particular, it is hard to get an intuition for the relationship between it and clustering. In particular, I am not sure what "trusted" means in this context? It is just a higher probability mass we're assigning to a certain sample?

1. Demšar, J. (2006). Statistical comparisons of classifiers over multiple data sets. The Journal of Machine learning research, 7, 1-30.

**Questions:**

1. You create a set of augmentations for an individual time series, why do you just pick one for training?
2. Can you confirm that all data sets have a train/val/test split and they were used accordingly for hyperparameter search?
3. Why not write $g_j^{Time}$ in Eq. 3?

---

### Comment · Reviewer_wHWj · 2024-11-15
**Updating Score**

After considering the work by Zhang et al., 2022 ("Self-Supervised Contrastive Pre-Training for Time Series via Time-Frequency Consistency"), I lower my score to 3: "reject, not good enough". While I still see some contributions beyond this paper, I agree with reviewer UJk1 that the way this work is referenced is insufficient and intransparent.

---

> ### Author Response · Authors · 2024-11-19
> **Urgent Response about the reference of paper [1].**
>
> We noticed that due to the similarity between our paper and [1], your rating dropped from 8 to 3, so we are urgently responding to you regarding this issue first.
>
> ### **1. About contribution**
>
> Our paper makes two key different contributions:
>
> 1. Quantifying the uncertainty of clustering results from both the time and frequency domains (Contribution 2).
> 2. Proposing an interactive clustering framework (Contribution 3).
>
> These contributions are entirely unrelated to [1]. Reviewer nLrp (rating: 6) recognized these as **''...novel directions...''**.
>
> ### **2. We treat [1] as a baseline**
>
> We included the method from [1], TFC, as a baseline for comparison (see Table 4 and Table 8). Our method significantly outperformed TFC across benchmark datasets on two common tasks—classification and anomaly detection.
>
> ### **3. Significant technical differences**
>
> Our similarity with [1] lies in the use of contrastive learning from both the time and frequency domains, which may lead you and Reviewer UJk1 to believe that we should heavily reference [1]. However, the essence of our work is fundamentally different. To clarify, we will explain this distinction through the construction of positive and negative pairs.
>
> Let $z_i^{time}$, $z_i^{freq}$, $\tilde{z}_i^{time}$, and $\tilde{z}_i^{freq}$ denote the embeddings (from time and frequency domains) of a sample $x_i^{time}$ and its augmentation $\tilde{x}_i^{time}$.
>
> - **Positive pairs in [1]**:
>   $(z_i^{time}, \tilde{z}_i^{time})$, $(z_i^{freq}, \tilde{z}_i^{freq})$, $\boldsymbol{(z_i^{time}, z_i^{freq})}$, $\boldsymbol{(\tilde{z}_i^{time}, \tilde{z}_i^{freq})}$
> - **Negative pairs in [1]**:
>   $(z_i^{time}, z_j^{time})$, $(\tilde{z}_i^{time}, \tilde{z}_j^{time})$, $(z_i^{freq}, z_j^{freq})$, $(\tilde{z}_i^{freq}, \tilde{z}_j^{freq})$
> - **Positive pairs in our paper**:
>   $(z_i^{time}, \tilde{z}_i^{time})$, $(z_i^{freq}, \tilde{z}_i^{freq})$
> - **Negative pairs in our paper**:
>   $(z_i^{time}, z_j^{time})$, $(\tilde{z}_i^{time}, \tilde{z}_j^{time})$,  $(z_i^{freq}, z_j^{freq})$, $(\tilde{z}_i^{freq}, \tilde{z}_j^{freq})$, $\boldsymbol{(z_i^{freq}, \tilde{z}_j^{freq})}$, $\boldsymbol{(z_i^{time}, \tilde{z}_j^{time})}$
>
> The bold part highlight where our approach diverges from [1], where [1] includes two additional positive pairs, and our paper includes two additional negative pairs.
>
> - In [1], additional positive pairs are constructed to enforce the model to **learn better embeddings based on the idea of ''time-frequency consistency''**. In this way, they fuse the information from time and frequency domains.
> - We fuse the results of FC layers from time and frequency domains, and **achieve better embeddings by ''enabling interactive learning between the fused clustering results and the pseudo-labels generated by k-means''**. This is fundamentally different from the core idea of [1].
> - We construct more negative pairs to enhance the separation between clusters, thereby achieving more distinguishable clustering results.
>
> ### **4. Different focus**
>
> The focus of our work is entirely different from [1]. While [1] studies pre-training for time-series models, we address time-series clustering.
>
> ### **5. Combining time- and frequency-domain contrastive learning is a very basic technique used in earlier papers.**
>
> It could be illogical to claim that our work originates from [1].
>
> - The use of data augmentation techniques in both time and frequency domains has been well-established, e.g., [2-3].
> - Combining time and frequency domains in representation learning existed prior to [1], as demonstrated by works in [4-6]. Does this mean that [1]'s method is not original?
>
> In [1], the authors claimed to be the first to apply frequency-domain data augmentation techniques in contrastive learning for solving the pre-training problem of time-series models. Similarly, we clearly stated in Contribution 1 that we are the first to apply frequency-domain data augmentation techniques to time-series clustering.
>
> **Reference**
>
> [1] Zhang X, Zhao Z, Tsiligkaridis T, et al. Self-supervised contrastive pre-training for time series via time-frequency consistency. *Neurips*, 2022.
>
> [2] Cui X, Goel V, Kingsbury B, et al. Data augmentation for deep convolutional neural network acoustic modeling. *ICASSP*, 2015.
>
> [3] Wen Q, Sun L, Yang F, et al. Time series data augmentation for deep learning: A survey. * arXiv:2002.12478*, 2020.
>
> [4] Liu D, Wang T, Liu S, et al. Contrastive self-supervised representation learning for sensing signals from the time-frequency perspective. *ICCCN*,2021.
>
> [5] Woo G, Liu C, Sahoo D, et al. CoST: Contrastive learning of disentangled seasonal-trend representations for time series forecasting. *ICLR*, 2022.
>
> [6] Yang L, Hong S, et al. Unsupervised time-series representation learning with iterative bilinear temporal-spectral fusion. *ICML*, 2022.
>
> We sincerely hope that you could restore your original rating. We will promptly address the other comments and make the necessary revisions.

---

> ### Author Response · Authors · 2024-11-19
>
> We are still working on your other comments. And we will respond to you as soon as possible.

---

> > ### Comment · Reviewer_wHWj · 2024-11-25
> >
> > 1. Your primary contribution claims to be "the first work to leverage contrastive augmentation in the frequency domain in a time-series clustering problem," which is reiterated in your response as "we are the first to apply frequency-domain data augmentation techniques to time-series clustering." However, this claim is demonstrably inaccurate, as evidenced by [1], which explicitly implements frequency-domain data augmentation techniques in time-series clustering contexts, stating "We also assess TF-C in extensive downstream tasks including clustering and anomaly detection."
> > 2. The assertion that adding 2 positive and negative samples constitutes a significant technical advancement appears to be overstated.
> > 3. With the first contribution invalidated, the second contribution requires substantially more rigorous analysis regarding clustering uncertainty quantification. While your approach incorporates distribution parameterization during the learning phase, the final cluster assignments are evaluated deterministically, without meaningful consideration of uncertainty. A more comprehensive comparison with existing uncertainty quantification methods in clustering would be necessary to substantiate this contribution.
> > 4. While the third contribution presents an interesting direction, its current scope and impact appear insufficient for the standards of this venue.
> >
> > Given these considerations, particularly the invalidation of the first contribution and the limited development of the second, I regret that the current work does not meet the threshold for acceptance at this venue.

---

### Note · Authors · 2024-11-26

I have read and agree with the venue's withdrawal policy on behalf of myself and my co-authors.